# Functional neuronal circuitry and oscillatory dynamics in human brain organoids

Tal Sharf [1,2,8] ✉, Tjitse van der Molen[1,2], Stella M. K. Glasauer [1,2], Elmer Guzman [1,2], Alessio P. Buccino [3], Gabriel Luna[1,2], Zhuowei Cheng[4], Morgane Audouard[1,2], Kamalini G. Ranasinghe[5], Kiwamu Kudo[6], Srikantan S. Nagarajan[6], Kenneth R. Tovar[1], Linda R. Petzold[4], Andreas Hierlemann [3], Paul K. Hansma[1,7] & Kenneth S. Kosik [1,2] ✉

Human brain organoids replicate much of the cellular diversity and developmental anatomy of the human brain. However, the physiology of neuronal circuits within organoids remains under-explored. With high-density CMOS microelectrode arrays and shank electrodes, we captured spontaneous extracellular activity from brain organoids derived from human induced pluripotent stem cells. We inferred functional connectivity from spike timing, revealing a large number of weak connections within a skeleton of significantly fewer strong connections. A benzodiazepine increased the uniformity of firing patterns and decreased the relative fraction of weakly connected edges. Our analysis of the local field potential demonstrate that brain organoids contain neuronal assemblies of sufficient size and functional connectivity to co-activate and generate field potentials from their collective transmembrane currents that phase-lock to spiking activity. These results point to the potential of brain organoids for the study of neuropsychiatric diseases, drug action, and the effects of external stimuli upon neuronal networks.

Under a variety of conditions[1], three-dimensional assemblies of human induced pluripotent stem cells (iPSCs), known as brain organoids, can differentiate into a wide diversity of brain cell types and assume anatomical features that resemble the developing brain as well as some mature features such as dendritic spines, inhibitory and excitatory synapses[2–4] with presynaptic vesicles[5]. Neurons within these three-dimensional assemblies generate action potentials upon depolarization[6], display excitatory and inhibitory post synaptic currents[7] and exhibit spontaneous network activity as measured by calcium imaging[6,8–10] and by extracellular field potential recordings from small numbers of electrodes[5,11–14]. Progress in the development of flexible electronics enable three-dimensional readouts of electrical activity distributed across the surface[15] and throughout the volume of an organoid[16], and recent work extended the activity repertoire of organoids to include rhythmic activity over a range of oscillatory frequencies[10,17].

Technological limitations have restricted acquisition of extracellular voltages to a relatively small numbers of neurons. Consequently, technical shortcomings have hindered attempts to determine functional connectivity among large neuronal networks, how the sum of their transmembrane conductances collectively synchronize with rhythmic activity and how pharmacologic perturbations affect these

[1]Neuroscience Research Institute, University of California Santa Barbara, Santa Barbara, CA 93106, USA. [2]Department of Molecular, Cellular and Developmental Biology, University of California Santa Barbara, Santa Barbara, CA 93106, USA. [3]Department of Biosystems Science and Engineering, ETH Zürich, Mattenstrasse 26, 4058 Basel, Switzerland. [4]Department of Computer Science, University of California Santa Barbara, Santa Barbara, CA 93106, USA. [5]Memory and Aging Center, Department of Neurology, University of California San Francisco, San Francisco, CA 94158, USA. [6]Department of Radiology and Biomedical Imaging, University of California San Francisco, San Francisco, CA 94143, USA. [7]Department of Physics, University of California Santa Barbara, Santa Barbara, CA 93106, USA. [8]Present address: Department of Biomolecular Engineering, University of California Santa Cruz, Santa Cruz, CA 95064, USA. ✉e-mail: tsharf@ucsc.edu; kosik@lifesci.ucsb.edu

physiologic parameters. Brain organoids provide many heuristic opportunities to reveal mesoscale features of neuronal signaling by linking single-unit activity to large network-level population dynamics across spatiotemporal scales impossible to resolve in vivo. The development of complementary metal-oxide-semiconductor (CMOS)-based microelectrode array (MEA) technology has enabled high-resolution readouts of extracellular field potentials generated by single neurons, at network scales, simultaneously over thousands of recording sites[18]. These technical innovations have revealed an unprecedented vantage of spatiotemporal dynamics governing propagating neuronal activity occurring in cortical-hippocampal circuits[19] and retinal network oscillations[20]. Continued advancements in CMOS-based MEA devices have increased the number of recording sites while retaining high signal-to-noise ratios[21], which have enabled large-scale ex vivo mapping of cortical synaptic projections[22], meanwhile the innovation of high-density three-dimensional electronic probes has opened the possibility of mapping neural activity in three-dimensions[23]. Here, we used state-of-the-art CMOS MEA technology to generate detailed electrical activity maps from 500 μm thick human brain organoid slices. These arrays have 26,400 routable electrodes, of which 1024 can be recorded from simultaneously. The surface area of these arrays (≈8 mm²) is comparable to the cross-sectional area of our brain organoids[24]. We constructed a network-level description of neuronal signaling in organoid slices by first constructing a spatial map of extracellular spikes generated by single neurons using densely tiled recording electrodes spaced at approximately the size of a neuronal soma (see Methods), thereby enabling robust and accurate assignment

of single-unit activity by utilizing electrode redundancy and the characteristic waveform shapes determined by the neuron's location relative to the recording electrode[25,26]. These technological advances offer a significant improvement over the more limited ascertainment of multi-unit activity (MUA) obtained from considerably sparser arrays used in previous studies. In addition, we used a 960 electrode Neuropixels CMOS shank probe[27] with an electrode pitch (≈20 μm) approximately the size of a neuron soma, to record from whole intact organoids. We quantified the probability distributions of single neuron spiking activity and found a subset of interspike intervals (ISIs) that are exponentially distributed having Poisson-like spike trains and another subset with more regular firing patterns. This emergent feature in human brain organoids resembles one facet of functional brain organization[28]. Benzodiazepine application shifted the ISI probability distributions toward lower frequency modes. These features were accompanied by less variation in burst to burst population-level dynamics. We constructed a spatial connectivity map derived from pairwise correlations between spiking units measured by the thousand microelectrodes selected from the total set of 26,400 electrodes with the highest activity. A subnetwork based on single-unit spikes demonstrated a marked increase in pairwise spike correlation strengths following benzodiazepine application. We detected theta frequency oscillations supported by their coherence across the surface of the organoid and stereotypical sets of neuronal ensembles phase-locked to the theta oscillation. The relationship between single-unit spikes and LFPs demonstrated here, as well as the stereotypical network behavior following benzodiazepine application suggest

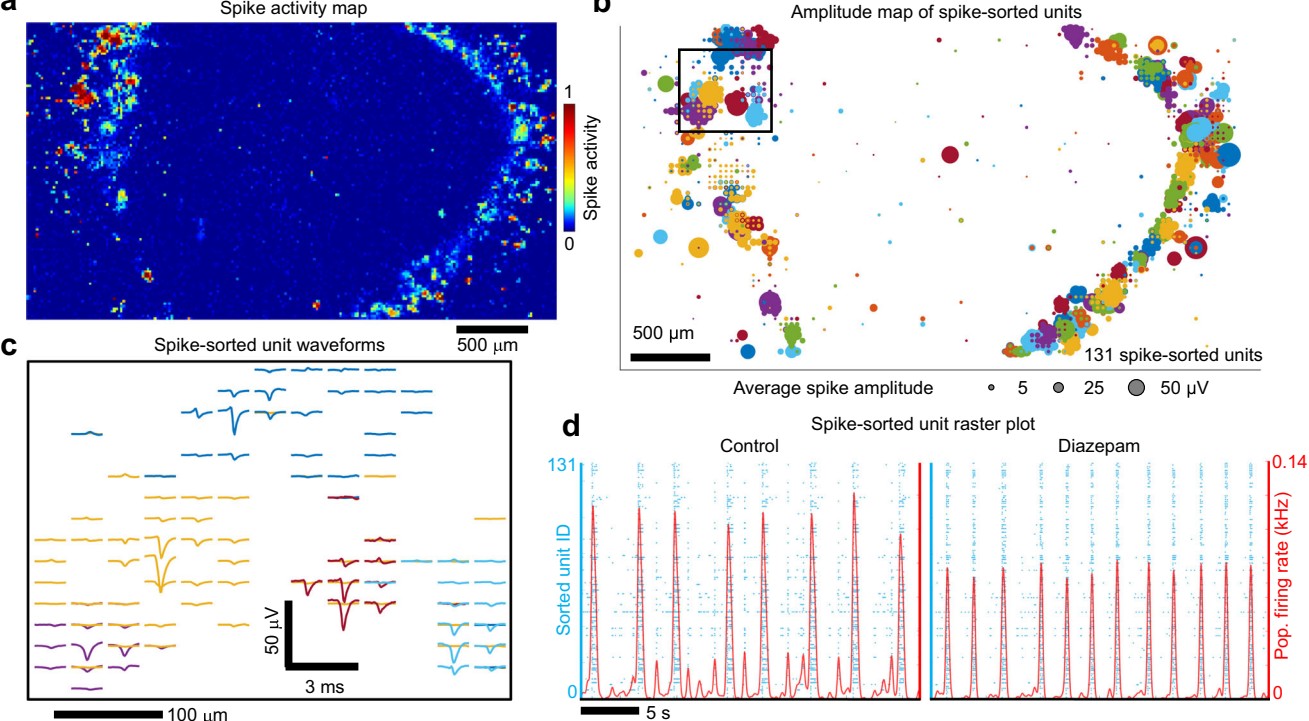

**Fig. 1 | High-resolution maps of extracellular action potentials across a human brain organoid slice. a** Spatial map of extracellular action potential activity recorded from a 500 μm thick human brain organoid slice positioned on a high-density CMOS microelectrode array with 26,400 recording electrodes to survey electrical activity across the entire organoid. The color scale indicates the normalized number of detected spikes (above 5x-rms noise) registered at each electrode site measured over a 30 s interval. Scale bar 500 μm. **b** Spatial map of the mean extracellular action potential spike amplitude (bubble size) from single-unit activity measured simultaneously across the CMOS array from the top 1020 electrode sites based on activity. A total of 131 spike-sorted units were determined by

Kilosort2. Single-unit electrode clusters are plotted using the same color (the same colors are repeated for different units). **c** Extracellular spike waveform traces from four individual units (from the region highlighted by the black square in **b**) are plotted with respect to the electrode positions on the array. The sorted unit colors are the same as **b**. Waveform scale bars are 50 μV and 3 ms; spatial scale bar is 100 μm. **d** Left, raster plot of endogenous spiking activity (blue dots) measured from the 131 spike-sorted units. Red line is the average number of spikes detected across the organoid (averaged over a 100 ms Gaussian kernel). Right, same organoid after 50 μM diazepam treatment. Source data are provided as a Source Data File.

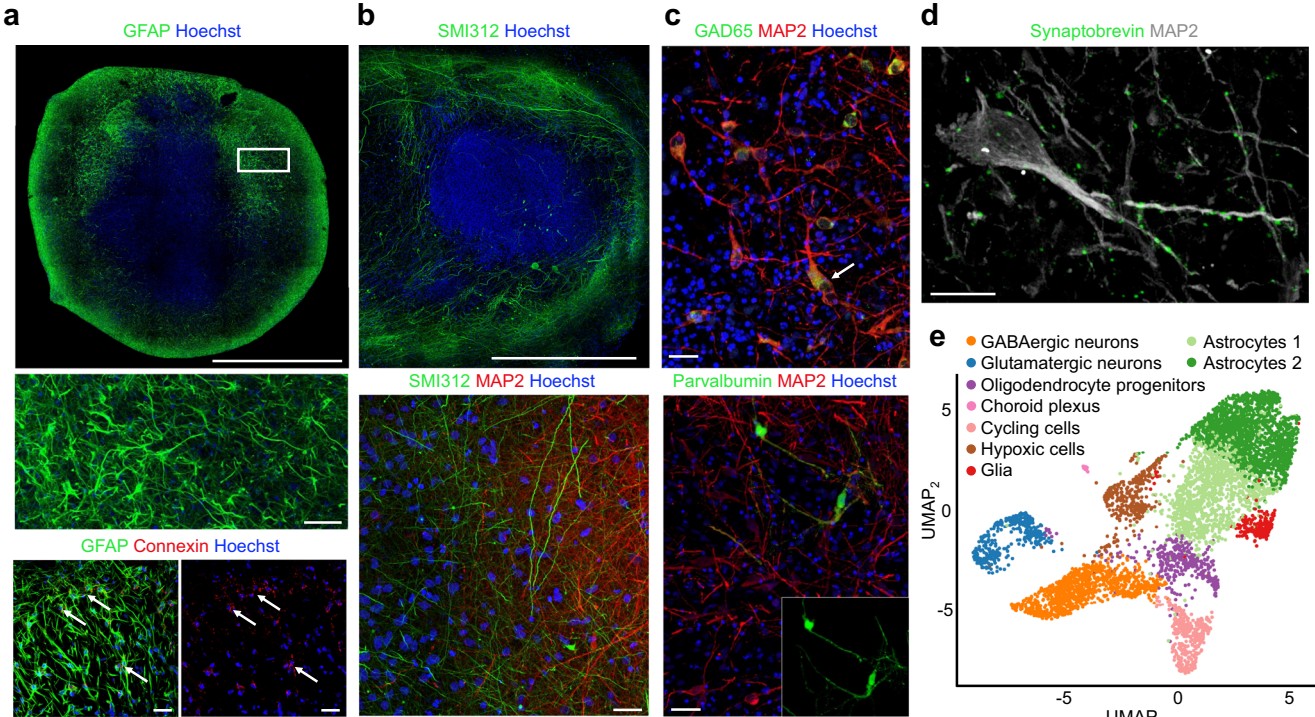

**Fig. 2 | Human brain organoids form a scaffolding capable of supporting neuronal microcircuitry. a** Top, high-resolution, whole-section of an 8-month organoid immunostained with anti-GFAP (green), and counterstained with Hoechst (cell nuclei, blue), scale bar 1 mm. Middle, a high magnification of (top box) showing stellate appearance characteristic of astrocytes. Bottom left, Anti-GFAP-positive astrocytes (green) in an 8-month organoid co-labeled with anti-connexin 43 demonstrating gap junctions (red). Bottom right, connexin 43 gap junction (red; arrows). Middle and bottom scale bars are 40 μm. **b** Top, long neuronal processes labeled with anti-SMI312 (green) in an 8-month organoid, scale bar 500 μm. Bottom, anti-SMI312 (green) axons neighboring MAP2-positive (red) neurons near the margin of 8-month organoid, scale bar 20 μm. **c** Top, anti-GAD65 positive neurons (green) co-labeled with anti-MAP2. Bottom, Anti-Parvalbumin-positive neurons (green) in an 8-month organoid co-labeled with anti-MAP2 (red). Scale bars 40 μm. **d** The presynaptic marker synaptobrevin label MAP2-positive processes as puncta, scale bar 20 μm. Data from **a–d** were repeated independently on $n = 3$ organoids. **e** Single-cell RNA sequencing (drop-seq) shows the presence of glutamatergic neurons, GABAergic neurons and astrocyte populations. Single-cell transcriptomes (5680 cells collected from three 7-month-old organoids) are visualized as a Uniform Manifold Approximation and Projection (UMAP). Source data are provided as a Source Data File.

ways organoids could be used to study a range of neurological conditions.

## Results

### High-resolution spike map across a human brain organoid slice

In order to acquire extracellular field recordings, 500 μm thick organoid slices were positioned over the recording electrode surface of high-density CMOS MEAs (MaxOne, Maxwell Biosystems, Zurich, Switzerland) and seated with a sterile custom harp slice grid for recording up to 6 months after placement on arrays (Supplementary Fig. 1, Methods). Slicing of organoids enhances cell growth, prevents interior cell death[12] and promotes extensive axon outgrowth[11]. In this study we present electrical activity from six organoids positioned on 2D CMOS arrays and performed pharmacological manipulations on a subset ($n = 4$) of that cohort. Additionally, extracellular recordings were performed on three whole organoids using CMOS shank electrodes. Once positioned on the MEA, spontaneous spiking occurred within ~2 weeks across organoids, followed by increasing firing rates in the form of synchronized bursts at about 6 months (Supplementary Data 1) and were maximally active at about 7 months (Supplementary Fig. 2a). Strong correlations between single-unit spike pairs emerged (Supplementary Fig. 2b, see section on short-latency interactions). A spike-activity map was derived by performing an automated scan of contiguous tiled blocks selected from 26,400 recording electrode sites using switch-matrix CMOS technology[24,29] that enabled configurable routing of up to 1024 simultaneous electrodes across the array

to on-chip readout electronics (Fig. 1a). We selected the top 1020 electrodes based on spiking frequency to record from and used these data to construct spatial maps of extracellular action potential waveforms that were used to identify single-units (Fig. 1b, c) using the spike-sorting algorithm Kilosort2[25], which has optimal accuracy and precision based on ground truth data for arrays with similar electrode densities[26]. Utilizing the single-unit spike times, which were often in the form of synchronized bursts (Fig. 1d), we generated detailed spatio-temporal maps of spiking activity across the organoid slice. To validate that spikes arose from fast synaptic transmission, we blocked AMPA and NMDA receptors with bath application of NBQX (10 μM) and R-CPP (20 μM) and GABA$_A$ receptors with gabazine (10 μM) resulting in a 72% ± 29% (mean ± STD) reduction in spiking (Supplementary Fig. 3; $n = 4$ organoids). The remaining action potentials in the absence of excitatory synaptic input could reflect interneuron firing[30]. A saturating concentration of the sodium-channel blocker tetrodotoxin (1 μM) was subsequently applied to the same organoid slices, resulting in a 98% ± 1% (mean ± STD) reduction in spiking activity compared to control (Supplementary Fig. 3), indicating that falsely detected spikes are a small fraction of the detected signals.

To understand the cellular basis of the physiology (Fig. 1), we characterized the cell types present in organoids, with particular attention to those cell types crucial for the formation, function and maintenance of networked neuronal circuitry[31,32]. We chose time points that coincided with the emergence of strong pairwise spike correlations in the organoids. Sections from 8-month old organoids

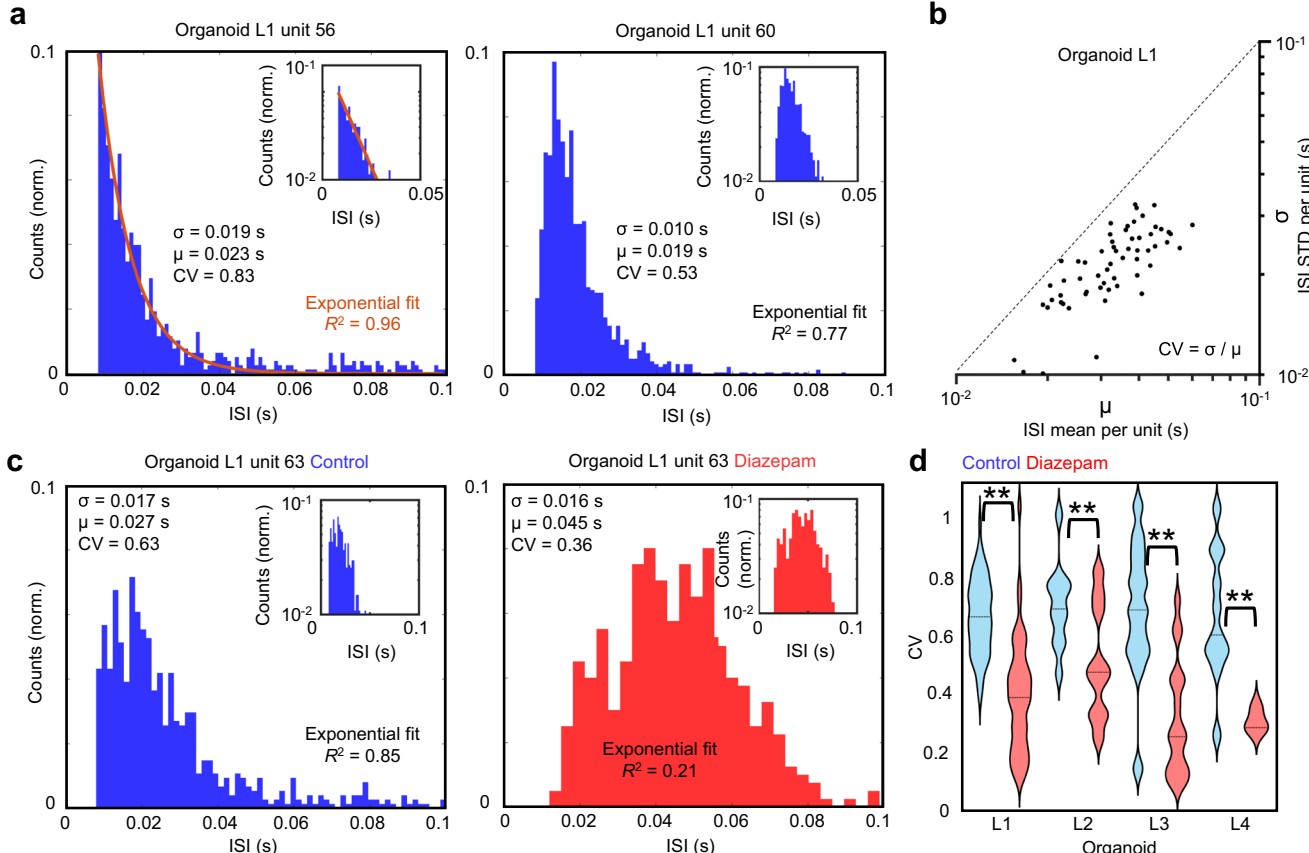

**Fig. 3 | Neuron interspike interval distributions from a human brain organoid slice. a** Histogram plots of interspike intervals (ISIs) from endogenous single-unit spiking activity are shown from organoid L1. The left plot shows an exponentially distributed ISI fit well by an exponential function (red line, $R^2 = 0.96$) with a coefficient of variation (CV), defined as the ratio of ISI standard deviation ($\sigma$) to its mean ($\mu$), close to one. The inset shows the vertical axis on a log scale. The right plot highlights a single-unit ISI with a right-skewed distribution, not as accurately captured by an exponential ($R^2 = 0.77$), with a comparatively smaller CV. **b** $\sigma$ is plotted as a function of $\mu$ across all units with a minimum of 30 spikes measured over a

3-min duration for organoid L1. The dotted line indicates CV = $\sigma/\mu = 1$. **c** Single-unit ISIs shift toward lower values with smaller CV for diazepam (50 μM) treatment relative to control. The distribution of all ISI intervals across all units is shown in Supplementary Fig. 8 for multiple organoids ($n = 4$). **d** The ISI CV distributions do not vary significantly between organoids in control conditions ($p > 0.4$). However, there is a significant reduction in the CV when treated with diazepam (50 μM) as determined by a two-sample Kolmogorov−Smirnov (KS) test ($p < 1e{-}4$, $p = 4.5e{-}2$, $p < 1e{-}4$, $p = 1.4e{-}2$ for organoids L1, L2, L3 and L4 respectively).

demonstrated the presence of stellate astrocytes expressing gap junction proteins (Fig. 2a). Axons coursed through dendritic fields that appeared directionally aligned in some regions and can project over long distances (Fig. 2b). Multiple interneuron types were present including parvalbumin-labeled cells (Fig. 2c)[7,33], as well as synaptobrevin decorated MAP2-positive dendrites (Fig. 2d). Due to diffusion limits within the organoid, we suspect that the interior of the organoid has reduced cell viability and is depleted of cells, thus accounting for the peripheral ring of higher spiking activity (Fig. 1a and Supplementary Fig. 4 for spiking activity from additional organoids). To obtain additional detail about the cellular composition of the organoids we performed single-cell sequencing on three 7-month-old brain organoids. Unsupervised clustering and dimensionality reduction resulted in the separation of nine major cell populations consisting of cortical excitatory neurons, GABAergic interneurons, astrocytes, an additional VIM+ glial cluster, oligodendrocyte progenitors, and a small cluster of choroid plexus cells (Fig. 2e and Supplementary Fig. 5). GABAergic cells expressed GAD1 and GAD2 (Supplementary Fig. 6) as well as several DLX genes similar to neurons that originate in the ventral forebrain (Supplementary Fig. 6). Within both neuronal clusters (glutamatergic and GABAergic), several subunits of GABA receptors associated with network oscillations[34] and AMPA and NMDA receptor genes were expressed (Supplementary Fig. 6). The forebrain marker FOXG1 was expressed in all these major cell populations. In contrast,

markers for other brain regions (retina, midbrain and hindbrain) were only minimally expressed (Supplementary Fig. 6), indicating a predominantly forebrain identity of our organoids[35].

These results indicate that the excitatory and inhibitory neurons present in our developing organoids are sufficient to result in self-organized, complex neuronal networks[2–8,11,12,17]. The presence of parvalbumin-positive GABAergic interneurons, a critical component for the function and timing of mammalian neuronal circuits[34], suggest that brain organoids contain functional circuit architecture necessary to generate a diverse mosaic of spiking patterns. Below, we quantify the spatiotemporal signatures of spikes and extracellular field potentials that arise from endogenous and pharmacologically induced activity in brain organoids.

## Neuronal spiking dynamics

Spike trains have been observed to follow Poisson-like dynamics in cortical circuits, where neurons fire action potentials with irregular ISIs that follow exponentially distributed probability densities[36–38]. However, a growing body of work has shown that a Poisson process of neuronal spiking is not a universal feature observed across cortical regions and subsets of neurons are contrasted by increased spiking regularity that reflect dynamics of brain-region specific circuitry[28,39,40]. To determine how spiking dynamics were distributed in human brain organoids, we generated ISI distributions from single-unit spike events

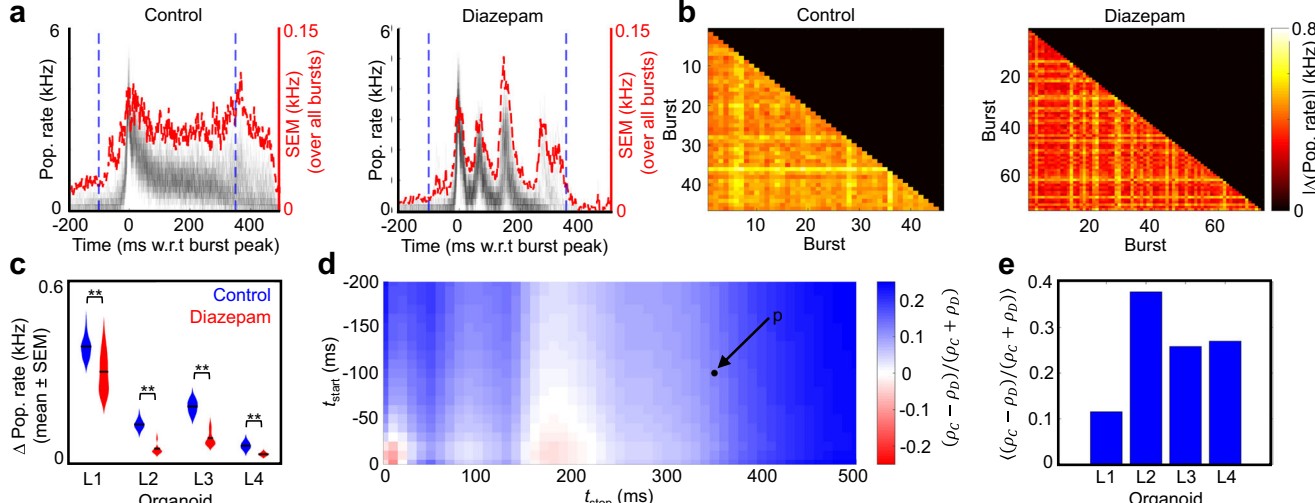

**Fig. 4 | Diazepam-induced changes in population-level dynamics in a human brain organoid slice. a** The population averaged firing rate (pop. rate) for the control and diazepam recordings (organoid L1 here) calculated from single-unit activity averaged over a 5 ms window. The population rate for each burst is plotted individually, centered by the peak in multi-unit activity (MUA). The standard error of the mean (SEM) calculated over all bursts is plotted in red. **b** Population rate average differences are shown across all individual burst pairs for the control and diazepam (50 μM) recordings from the same organoid. The average population rate difference is computed over a time window of −100 ms to +350 ms relative to the MUA peak (blue dotted line in **a**). The color bar scale is the same for control and diazepam. See Supplementary Fig. 10 for visualization from a separate organoid. **c** Average population vector differences taken over all individual burst pairs per recording for control and diazepam (50 μM) conditions. The population vector

difference is computed over a time window of −100 ms to +350 ms relative to the MUA peak (blue dotted line in **a**). Significance between control and diazepam was determined by a two-sample KS test ($p < 1e{-}4$ for organoid L1, L2, L3 and L4). The number of pairwise burst comparison for each distribution for control and diazepam (50 μM) conditions, respectively, are (1035, 2701); (5460, 10,153); (3570, 7875); (1275, 861) for organoids L1, L2, L3 and L4 respectively. **d** Fractional change between population rate for control and diazepam $(\rho_C - \rho_D)/(\rho_C + \rho_D)$, averaged over a range of different time windows for L1. Here, $\rho_C$ and $\rho_C$ are the average population rate differences for control and diazepam, respectively. Blue tiles indicate a higher average population vector difference for control. The black dot ($p = (-100, 350$ ms)) indicates the time window used in **b** and **c**. **e** Average score of the matrix in **d** for each organoid.

as shown in Fig. 3a (see "Methods" for additional details). We observed a subset of units with exponentially distributed ISIs, shown by the red line in Fig. 3a (left) that accounted for $16\% \pm 8\%$ from a set of 224 units identified across multiple organoid recordings ($n = 4$ organoids) with a stringent $R^2$-threshold set to 0.9. We next utilized the coefficient of variation (CV), defined as the ratio of the standard deviation ($\sigma$) in the ISI to its mean ($\mu$) to map the variability of our measured ISIs (Fig. 3b). ISIs following a perfect Poisson process have a $CV = 1$[37], whereas a perfectly periodic spike train will have a CV near zero[28]. Together, these results suggest that non-random spiking dynamics, which are considered fundamental to functional and structural brain organization are emergent features in human brain organoids.

### Diazepam reduces spiking complexity within brain organoid neuronal circuits

Diazepam, a benzodiazepine, is a commonly used anxiolytic-hypnotic drug with a range of therapeutic effects on the central nervous system that range from sedation at low doses to induction of anesthesia at considerably higher doses[41–43]. We investigated diazepam action on spiking behavior in organoid neuronal networks. Diazepam (50 μM) decreased the inter-burst intervals, burst duration and the number of spikes participating in the burst (burst amplitude) throughout the organoid ($n = 4$ organoids, Fig. 1d and Supplementary Fig. 7 for diazepam dose responses over the range from 3 to 50 μM and additional statistics), consistent with diazepam effects in neocortical slice cultures over the same concentration range[44].

Treatment with diazepam (50 μM) increased the distribution of ISIs toward longer intervals (Fig. 3c) as observed across multiple organoids ($n = 4$, Supplementary Fig. 8), and a reduced ISI variability as determined by the CV relative to control (Fig. 3d) (two-sample Kolmogorov–Smirnov (KS) test ($n = 4$, $p < 0.05$)). Furthermore, upon the addition of diazepam (50 μM) the of ISIs fraction with a $CV > 0.7$

drops to $8\% \pm 10\%$ ($n = 4$ organoids) from $42\% \pm 4\%$ of the units under control conditions. Interestingly, the total number of spikes per unit remained similar ($n = 4$ organoid slices, Supplementary Fig. 9).

Concomitant changes in single-unit spiking activity were observed at the population level (Fig. 4). Temporal dynamics of coordinated neuronal bursting activity are visualized by the population averaged firing rate (averaged over a 5 ms Gaussian kernel, see "Methods") for all bursts relative to their peak onset times (Fig. 4a, gray lines). Burst onsets initiate within 100 ms and persist for several hundred milliseconds before subsiding. Variation in burst-to-burst activity is visualized by the standard error of the mean (SEM) across all burst events (Fig. 4a, red line). For control conditions the SEM typically peaks during the burst onset and reaches a steady state value during the duration of the burst. However, for diazepam (50 μM) the burst-to-burst SEM remains, on average, at a lower value. Figure 4b highlights the average difference in population firing rates (over the blue dotted intervals in Fig. 4a) across all burst events in control or 50 μM diazepam treatment. Interestingly, diazepam significantly reduced variation in the temporal structure of population-level activity across organoids (Fig. 4c) when averaging over all burst epochs (two-sample KS test, $p < 1e{-}4$).

To further dissect variation in burst-to-burst activity, we examined the fractional change in the average population rates over a given time interval relative to the multi-unit activity (MUA) burst peak (Fig. 4d), given by the following expression $(\rho_C - \rho_D)/(\rho_C + \rho_D)$. Here, $\rho_C$ and $\rho_D$ are the average population rate differences for control and 50 μM diazepam conditions, respectively, for the given time interval relative to the burst peak set by $t_{start}$ and $t_{stop}$. Overall, we observed increased burst-to-burst variability when comparing control conditions relative to diazepam (across the full burst window); however, there was a slight increase in diazepam variability for organoid L1 over the narrow interval [150, 200] ms relative to the

burst peak (this increase was not present in other organoids, see for example Supplementary Fig. 10). When averaging over all burst intervals (all matrix elements shown in Fig. 4d) we observed a consistent increase in burst-burst variability across organoids for control conditions relative to diazepam (50 μM) treatment (Fig. 4e). These observations, combined with a significant decrease in the ISI CV (Fig. 3d), suggested that diazepam skews neuronal spiking in our organoid slices toward a more redundant and strongly interconnected network with the property of instantiating active functional states.

## Short-latency interactions in human brain organoid slices

Functional couplings between neurons would be expected to have latencies of ≈5 ms, to have pairwise spike correlations less than one due to the probabilistic nature of synaptic transmission[45,46], and to be inherently nosier than high-fidelity action potentials[47,48]. Quantification of functional couplings between neurons according to these criteria allowed us to derive a map of pairwise spike correlations between single-unit spikes and their latency distributions (Fig. 5a, b). The distribution of pairwise mean spike latencies was $4.7 \pm 2.3$ ms measured across six organoids from 14 separate measurements, as observed in neocortical circuits[45]. To quantify the correlation strength between units, we computed pairwise spike correlations using the spike time tiling coefficient (STTC) (Fig. 5b inset), which takes into account neuron firing rates that can confound traditional cross-correlogram approaches[49] and generates weighted pairwise spike correlations that we refer to as edges (Fig. 5c). Edge strengths were determined by the magnitude of the STCC (Fig. 5c, line thickness) using a correlation window size of $\Delta t = 20$ ms. The correlation values reached a stable plateau for $\Delta t \geq 20$ ms (Fig. 5b inset), consistent with the histogram (Fig. 5b). The relative delay times between putative connections identified by correlated spike trains and spike-time jitter resulted in a mean latency within ≈5 ms time window with very few latencies longer than 10 ms (see Supplementary Fig. 11a for latency distributions from multiple organoids). These observations demonstrate that pairwise spike correlations emerge from spontaneous, network driven, spiking activity at timescales consistent with synaptic transmission.

Directionality was assigned to the edges of the pairwise spike correlation map based on the spike latency distributions between single-unit pairs (see "Methods"). A negative mean spike time latency compared to the reference unit indicated a predominant directionality toward the reference unit and vice versa (Fig. 5b). Bi-directional connections as suggested by non-unimodal latency distributions were excluded from the analysis and constituted a small fraction of the total identified connections ($7.13\% \pm 5.06\%$ $n = 14$ measurements across $n = 6$ organoids). Based on the directionality, the in-degree ($D_{in}$) and out-degree ($D_{out}$) were computed for each unit, defined as the number of incoming and outgoing edges respectively. Nodes with a high fraction of outgoing edges ($(D_{out} - D_{in})/(D_{out} + D_{in}) > 0.8$) were labeled "sender" nodes (Fig. 5d, top). Nodes with a high fraction of incoming edges $(D_{in} - D_{out})/(D_{out} + D_{in}) > 0.8$ were labeled "receiver" nodes (Fig. 5d, middle). Differences in the directionality vector <0.8 were labeled "broker" nodes (Fig. 5d, bottom), which represented the majority of the nodes (senders: $15.4\% \pm 2.6\%$, brokers: $62.7\% \pm 6.0\%$, receivers: $21.9\% \pm 6.5\%$). These data were computed from four independent organoids (Supplementary Fig. 12a, b).

The assembled pairwise spike time correlation map revealed a directional traffic pattern through the organoid (Fig. 5d). The largest component—a graph of contiguous interconnected nodes—was parameterized according to its edge strengths as determined by the magnitude of pairwise spike correlations between spike trains. Setting the lower STTC limit to 0.35 minimized spurious connections (see "Methods" for randomization details and Supplementary Fig. 11b), and the number of active units (nodes) and global edge weight distributions were stable (>4 h) in that they did not vary significantly under

control conditions (two-sample KS test; $p \geq 0.1$, Supplementary Fig. 13a, b). Moreover, the global distribution of edge strengths did not vary significantly over minute-long timescales (two-sample KS test; $p \geq 0.3$, Supplementary Fig. 13c). Edge strengths did not fit a random distribution (Fig. 6a and Supplementary Fig. 14). Rather, a truncated power law[50] and a gamma distribution fit experimental data well, with the gamma distribution outperforming the truncated power law (Methods and Supplementary Fig. 14). These findings support the hypothesis that neuronal circuitry in brain organoids is non-random and composed of a minority population of strong connections, reflecting the anatomical non-random circuitry strength found in the visual[51] and primary somatosensory cortex[52].

To assess the modifiability of these organoid network maps, we increased the inhibitory tone by the addition of diazepam (50 μM) which increased the relative fraction of strong edges while decreasing the relative abundance of weaker ones (Fig. 6a). The change in the edge strengths was further visualized as a fractional difference $(f_d - f_c)/(f_d + f_c)$, where $f_c$ and $f_d$ are the fraction of total edges for control and diazepam (50 μM) respectively ($n = 3$ organoids, Fig. 6b). Diazepam eliminated a population of weakly connected edges (Fig. 6c), while inducing a more strongly interconnected subnetwork (Fig. 6c–e). These results combined with an increase in the density of edges ($40\% \pm 11\%$, $n = 3$ organoids) reveal a more correlated and interconnected network with less variable single-unit and population-level neuronal dynamics in the presence of diazepam (Supplementary Fig. 12c).

## Theta frequency oscillations

The transmembrane currents associated with action potentials are summed in local networks and are detectable as local field potentials (LFPs). Additionally, extracellular voltage fluctuations associated with synaptic transmission are thought to represent the coordinated activation of neuronal ensembles and occur within frequency domains characteristic of many LFPs[53]. We examined the relationship between single-unit spiking, MUA and LFPs at the theta frequency in organoid slices.

We used band-pass filtering of raw extracellular voltage data to extract LFPs (Fig. 7a) and observed multiple sub-band frequencies (Supplementary Fig. 15). We limited our analysis to theta oscillations (4–8 Hz) that were significant over time intervals when the oscillation amplitude envelope exceeded the noise floor (Supplementary Figs. 16 and 17). Local theta cycles (Fig. 7b, red lines) were referenced to burst activity (Fig. 7b, black line) signifying when the MUA population-averaged spike rate peaked within the burst. Theta oscillations showed consistent phase offsets relative to each other (Fig. 7c, gray lines). When we blocked AMPA, NMDA and GABA$_A$ receptors along with TTX-sensitive sodium channels, the LFP amplitudes were no longer detected ($n = 4$ organoids; Supplementary Fig. 16) validating the conclusion that the dominant source of LFPs arose from network activity driven by synaptic transmission[53].

Cross-correlation coefficients of theta oscillations and their relative time-lags between all electrode pairs were used to generate a spatial correlation map with a focal region of highest correlated activity and their phase delays relative to more distant positions on the map (Fig. 7d). The spatiotemporal evolution of theta oscillation amplitudes is visualized during the onset of a neuronal population burst in Supplementary Fig. 18a, further illustrating non-stationary aspects of LFPs generated by network activity in our organoids. Interestingly, non-stationary features of the LFP have also been observed in brain tissue preparations with complex neuronal architectures such as cortico-hippocampal[19] and retinal[20] networks using high-density CMOS MEAs. Signal averaging theta oscillations with respect to oscillation peaks from the node with the highest correlation (identified in Fig. 7d) further validated synchronized theta activity with relative phase shifts across the organoid (Fig. 7e, f and see

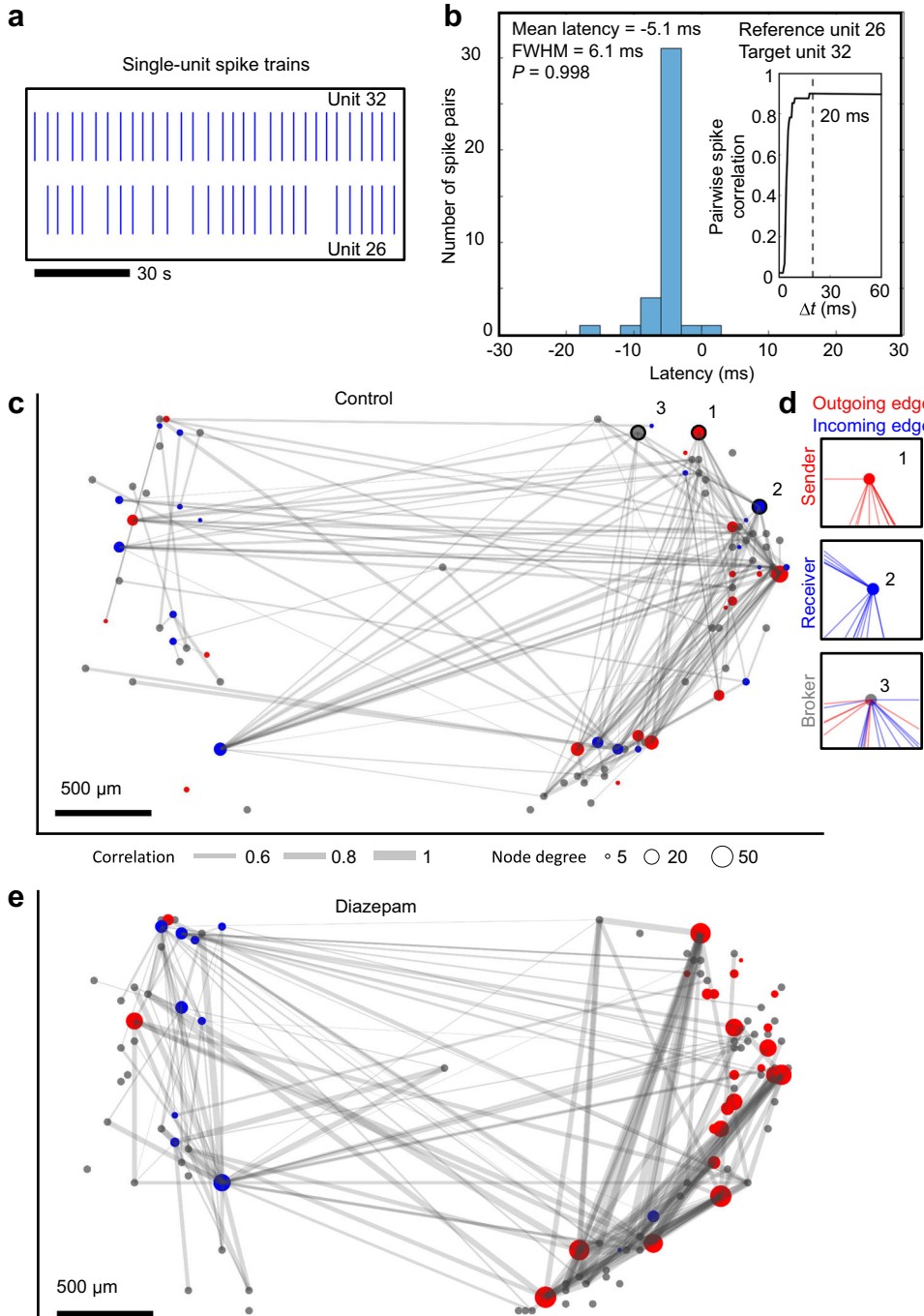

**Fig. 5 | Mapping information flow through a human brain organoid slice.**
**a** Extracellular action potential spike events are shown from two correlated spike-sorted units. **b** The spike time latency distribution is shown between the two spike-sorted units shown in **a**. The latency distribution between pairwise spike events is unimodal (Hartigans' dip test for multimodality, $p = 0.998$), with a mean latency of 5.1 ms and a FWHM of 6.1 ms. The inset shows the pairwise spike correlation was calculated using the spike time tile coefficient (STTC) as a function of correlation time window ($\Delta t$) for the two correlated spike-sorted units shown in **a**. Choosing a correlation time window $\Delta t = 20$ ms captures all pairwise spike interactions between the two units. **c** Functional connectivity map showing the pairwise correlation strength (edge thickness in gray) between spike trains. Sorted by directionality, the in-degree and out-degree were computed per unit, defined as predominately incoming or outgoing edges respectively and designated "receiver" (blue) nodes, "sender" (red) nodes. All other nodes were labeled "brokers" (gray with a fixed size not indicative of node degree). For visual clarity, only the top 90 outgoing and the top 90 incoming edges are shown for "sender" and "receiver" nodes, respectively. **d** Examples of single "sender" (1), "receiver" (2) and "broker" (3) nodes showing all incoming (blue) and outgoing (red) edges for the spatial sites identified on **c**. The relative fraction of sender, receiver and broker edges remained similar across multiple organoids ($n = 4$, Supplementary Fig. 12a, b). **e** Functional connectivity map of the same organoid after treatment with 50 μM diazepam.

Supplementary Fig. 18b–h for visualizations in a different organoid). We confirmed the validity of the LFP theta signal by estimating imaginary coherence, a metric which depends on non-zero phase synchrony[54], and found sets of highly coherent electrodes (Supplementary Fig. 19a–e). We next used K-means clustering to identify a region with significantly higher coherence relative to the other spatial locations across the organoid (Wilcoxon ranked sum test, $p < 1e-4$) and found a prominent topological overlap with electrode

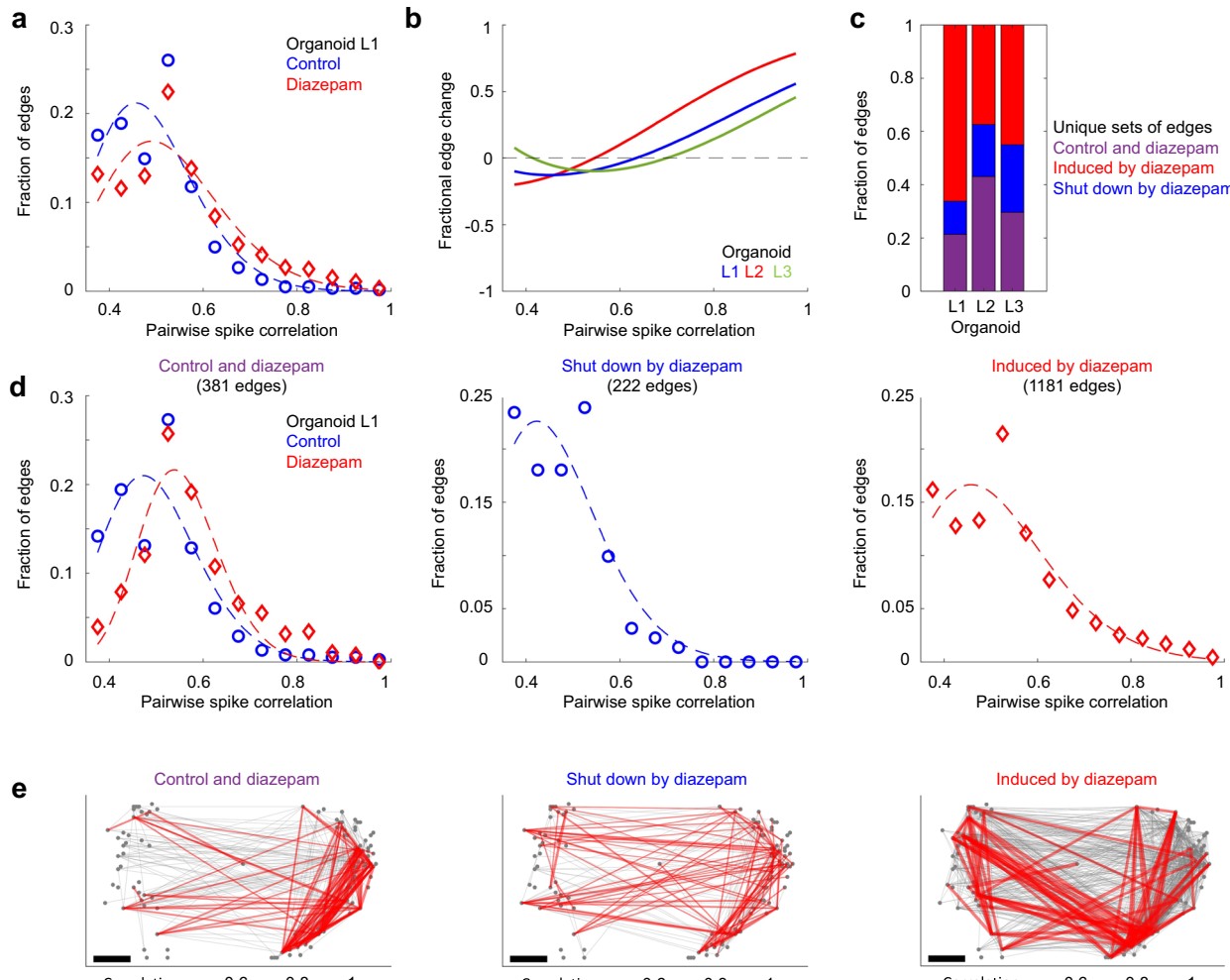

**Fig. 6 | Diazepam selectively modulates the skeleton network of strong edges.** **a** Pairwise spike correlation (edge) distributions are shown from the spatial maps of Fig. 5c, e. The edge strengths (of the largest component of interconnected nodes) were binned using a bin size = 0.05 over a connectivity range from 0.35 to 1. A gamma distribution fit illustrates "a skeleton of stronger connections immersed in a sea of weaker ones". Diazepam (50 μM) increased the minority population of stronger edge strengths, while decreasing the strength of the more abundant weaker edges with respect to control conditions. **b** The fractional difference in edge strengths, $(f_d - f_c)/(f_d + f_c)$, is shown, where $f_c$ and $f_d$ are the gamma distribution fits highlighted in **a** for control and diazepam (50 μM) conditions, respectively. A higher proportion of strong edges are present in diazepam (50 μM) conditions relative to control conditions for $n = 3$ organoids. **c** The relative fractions of unique sets of edges are shown. Purple bars represent edges present during both control and diazepam conditions. Blue bars represent the fraction of edges shut down by diazepam, and the red bars represent the fraction of edges induced by diazepam. **d** Edge strength distributions highlighted in **c** are plotted for organoid L1, while the spatial graph is shown in **e**. The top 90 edges (by weight) are shown in red, while the remaining edges are plotted in gray.

---

sites identified by cross-correlation analysis (Supplementary Fig. 19g, h), resulting in a 44% overlap of the electrode sites. Furthermore, diazepam induced a dramatic reduction in pairwise theta correlations (Supplementary Fig. 20d) and a significant decrease in imaginary coherence across the organoid ($n = 3$ organoids, Wilcoxon ranked sum test, $p < 1e{-}4$, Supplementary Fig. 19f).

### Peak theta amplitudes are synchronized with neuronal population bursts

Spatiotemporal synchronization of action potentials in local neuronal populations generate oscillatory rhythms observed in the LFP[53]. The amplitudes of these oscillations are correlated with an increase in local population-level spiking activity, as observed in the visual[55] and primary motor cortex[56] and elsewhere. We also observed a sharp increase in the amplitude of theta oscillations occurring during time periods exhibiting synchronized neuronal population bursts (Supplementary Fig. 17). First, we identified when the neuronal population firing rates (MUA averaged over a 5 ms time window) peaked within burst epochs.

Burst-peak times were then used as temporal anchor points to signal average theta oscillations (Fig. 7g, h) across all electrodes. A spatial map (Fig. 7i) of signal-averaged theta oscillations (relative to the MUA population burst-peak) revealed regions of theta activity strongly correlated with burst peaks. Signal averaging theta amplitudes relative to population burst-peaks revealed the same spatial pattern of theta phase coherence attained by independently signal averaging theta oscillations relative to their peak oscillation amplitude time points (Fig. 7e, f). Theta phase alignment was maximal within 100 ms after the population burst peak (red dotted line is the burst peak in Fig. 7g and inset) and shown by a sharp drop in the circular standard deviation of the phase (phase angle spread). Temporal windows of constant phase shifts (defined where the circular standard deviation of the theta phase dropped below one) persisted for 2 to 3 theta cycles (≈400 ms). The time above the noise floor substantially increased during broader burst periods ($n = 4$ organoids, Supplementary Fig. 17). The addition of diazepam (50 μM) reduced the number of electrodes correlated at theta frequencies (above a correlation threshold of 0.2) across the

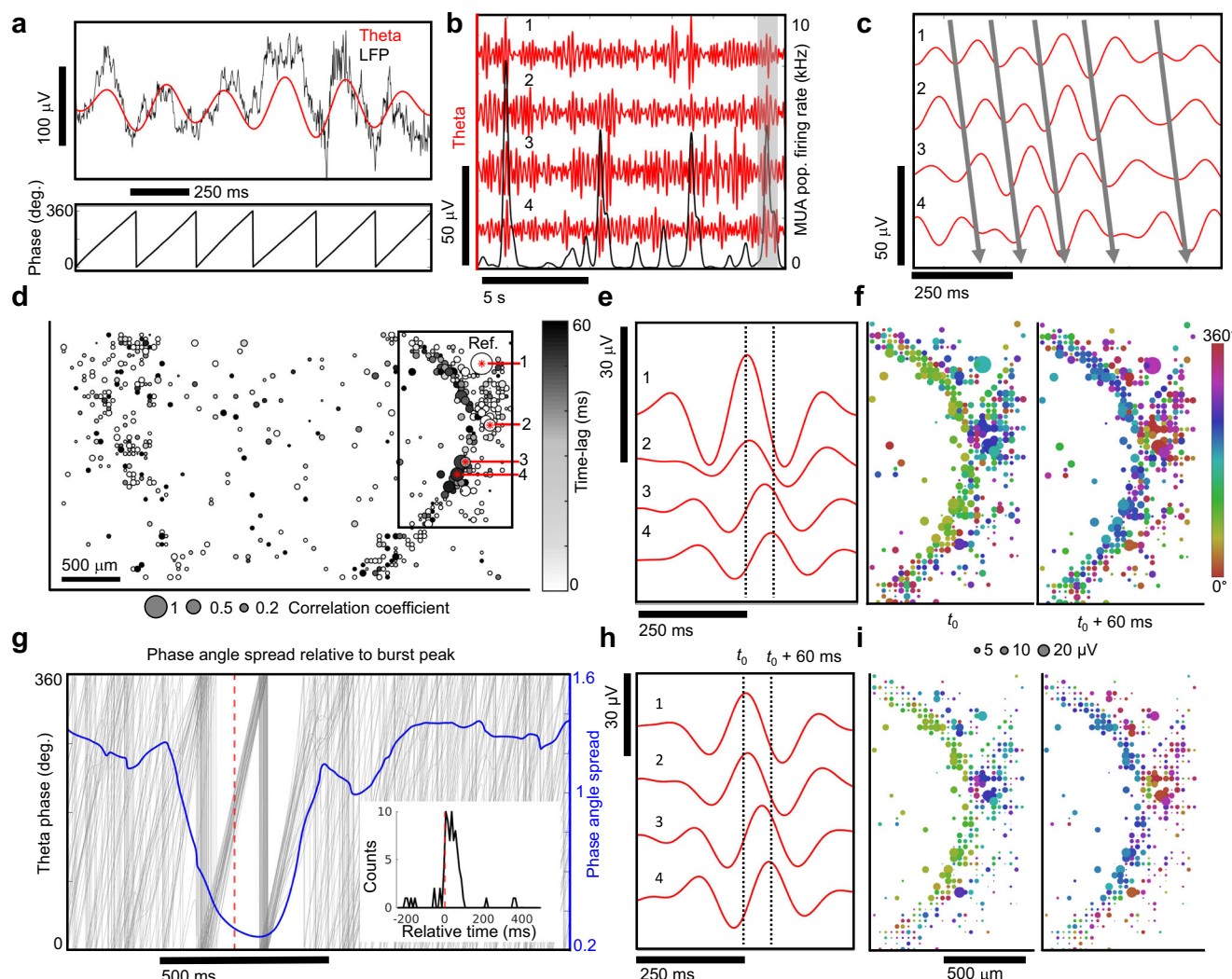

**Fig. 7 | Spatial and temporal coherence of theta oscillations with neuronal population bursts. a** The raw local field potential (<500 Hz, black line) and the 4−8 Hz theta filtered band (red) (top). The phase of the theta oscillation (bottom). **b** Theta band oscillations (red) from four different recording sites and the multi-unit activity (MUA) population averaged firing rate (black) averaged over a 100 ms window. **c** Zoomed in view of highlighted black rectangle in **b**. The solid lines indicate the relative phase offsets of theta oscillations across spatial sites of the organoid. Within a narrow time window these oscillations showed consistent phase offsets. **d** Spatial correlation map of theta oscillations. The correlation coefficient (bubble size) is shown with respect to the seed reference site (1) and the relative phase-lag with respect to the reference electrode shown in grayscale reveals spatial alignment of theta oscillations. **e** Signal averaged theta oscillations using peaks from electrode (1) as a reference. The numbers 1–4 in **c**−**e**, **h** all refer to the same set of electrodes. **f** Spatial map of signal averaged theta oscillation phase and amplitude relative to reference electrode number 1. Two time points are shown, one at the center of the reference electrode $t_0$ and another 60 ms later. **g** Phase angle spread in radians (blue line) is plotted relative to the burst peak. Individual theta phase traces (gray line) are plotted for electrode 1 relative to population burst events determined from MUA averaged over a 5 ms window. The phase angle spread is minimized after the burst peak (red dotted line). The time of the theta peak amplitude relative to bursts where the angular spread is minimized across multiple electrode sites (inset). **h** The same theta oscillations from **e** are signal averaged with respect to population burst peak times ($t_0$). **i** Spatial map of signal averaged theta oscillations using population burst peak times reveal a temporal alignment of theta oscillations with neuronal population bursts. **f**, **i** share the same scales.

organoid slice ($n=3$) (Supplementary Fig. 20d), a result consistent with down-regulation of theta activity by benzodiazepines as seen in human cortical EEG[57] and intracranial recordings from bipolar electrodes placed in the frontal, parietal and occipital lobes of the murine brain[58]. Taken together, our results show that theta oscillation amplitudes are correlated with population spiking activity and provide a platform for further understanding the integration of local neuronal activity with summed LFPs observed in human EEG and MEG recordings[55].

## Spike phase-locking to theta

In the absence of sensory input, brain activity can entrain action potentials to theta frequency oscillations[59] and define a temporal window for spiking within local circuits[60,61]. Previous work in vivo and

ex vivo has relied on manually positioned, low-density recording electrodes, to identify a handful of units exhibiting preferential spike phase-locking to theta frequencies at a given moment in time[59,60,62,63]. Advancements in CMOS-based MEA technologies have enabled the possibility to establish quantitative relationships emerging among LFPs and spiking dynamics at the single-cell level across full slice networks[19], map large-scale retinal network oscillations[20] and more recently used to uncover large-scale coherence patterns due to changes in behavioral states in the monkey cortex[64]. We looked for relationships between theta frequency oscillations and single-unit spikes in organoids using the high-density and large spatial extent of the CMOS arrays. To determine whether spikes occurred at a preferred theta phase, we counted spikes in relation to theta cycle phase (Fig. 8a) and determined significance with the Rayleigh criterion for non-

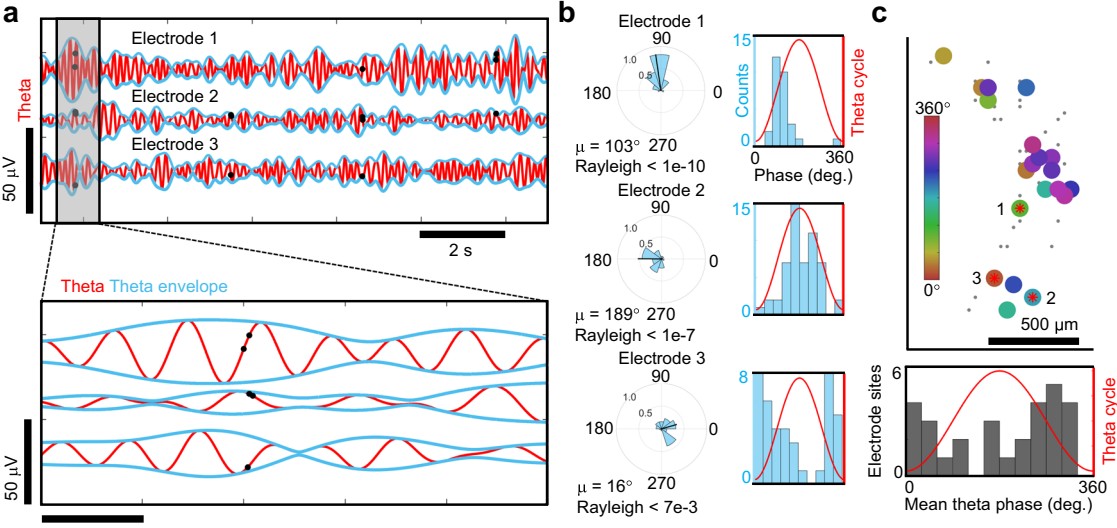

**Fig. 8 | Phase-locking of spikes to theta oscillations. a** Top, theta oscillations (red line) from three representative electrode sites and the single-unit spikes that occurred at each of those electrodes (black dots). Bottom, zoomed in view from the gray box on the top. **b** Left, circular distribution of theta phase angles occurring during single-unit spike events measured from the electrodes in **a**. Direction of the mean spike angle (μ) relative to the theta phase and magnitude (mean resultant length) are shown in the polar plots. The Rayleigh test for non-uniformity was used to determine if spikes were distributed non-uniformly over the theta cycle (0°, 360°). Right, distributions of theta-spike angles shown relative to the theta cycle are visualized as histogram plots. **c** Top, a cluster of phase-locked units to theta oscillations are shown within the coherent pocket highlighted by the box in Fig. 7d. The color indicates the mean phase-locked angle to theta (μ) that satisfy the Rayleigh test for non-uniformity ($p < 0.05$). A subset of the total electrode sites with no preferred theta phase ($n = 102$, $p > 0.05$) are shown by gray dots. Bottom, histogram of the mean phased-locked angle to theta for all electrode sites across the array that satisfy the Rayleigh criteria ($n = 29$, $p < 0.05$) which account for 22% of the 131 total active units detected. We observe an average of 28% ± 14% of single units exhibiting phase-locking to theta oscillations based on the Rayleigh criteria defined above ($n = 4$ organoids at 8 months).

uniform distribution of spikes in circular phase space[59]. By selecting the most active 1020 microelectrodes (out of 26,400), we simultaneously surveyed theta oscillations and spikes across the organoid slice to construct a phase-locking map for regions of highly correlated theta activity identified previously by the cross-correlation analysis (Fig. 8c and Supplementary Fig. 21 for plots from additional organoids). Theta-phase spike event distributions for electrode locations revealed a strong preference for spiking during a narrow phase window of theta centered about its mean angle μ (Fig. 8b). Interestingly, we found that 28% ± 14% of single-units ($n = 4$ organoids) exhibit phase-locking to theta frequency oscillations in our organoids within the Rayleigh criteria for non-uniformity ($p < 0.05$), other frequency bands, such as gamma, were not investigated and could account for other distinct populations reported elsewhere that exhibit phase-locking in these frequency regimes[65]. Theta oscillations provide a temporal window for computation within local circuits in situ[60,66], suggesting that organoids may display sufficiently complex circuitry for intrinsic computation.

### Extracellular recordings in whole organoids

To determine if the spiking characteristics in organoid slices are recapitulated in whole organoids, we recorded extracellularly from intact brain organoids, using a densely tiled 960 electrode CMOS shank with ≈20 μm inter-electrode pitch (Neuropixels probe)[27]. Recordings were made by inserting the shank into the organoid using a motorized, low-drift micromanipulator. We measured single-unit spiking across hundreds of microns of organoid tissue within the z-plane in whole organoids ($n = 3$; Fig. 9a, b). As in organoid slices, we observed spontaneous, synchronized population bursts in recordings from whole organoids.

We confirmed synchronization of theta oscillations to neuronal population bursts and identified electrode sites along the z-plane of whole organoids that exhibited prominent theta band oscillations in the LFP (Fig. 9c). Population bursts were phase-synchronized to theta oscillations in the intact organoids, as determined by signal-averaging theta phases relative to MUA population burst peak times along the shank (Fig. 9d, e) and co-localized with spatial regions where spiking activity was observed. Theta phases added deconstructively in organoid regions showing no spiking (Fig. 9e, right). Our measurements of electrical activity generated by brain organoids provided strong evidence that theta is synchronized over temporal epochs of increased neuronal activity within the time window of population bursts[53]. Finally, we confirmed phase-locking of spikes to theta across multiple sites in whole brain organoids (Fig. 9f, g). The presence of spiking and LFP activity verified that network activity occurred within a stratum several hundred microns within the three-dimensional structure of the organoid and paralleled the activity observed from slices on planar high-density arrays.

## Discussion

Human brain organoids are an intrinsically self-organized neuronal ensemble grown from three-dimensional assemblies of human-iPSCs. As shown here, brain organoids offer a window into the complex neuronal activity that emerges from intrinsically-formed circuits capable of mirroring aspects of the developing human brain[32]. Applying high-density CMOS MEA to large multi-cellular networks spanning millimeters of the brain organoid cross-sections we isolated single-unit activity and computed the timing of successive action potentials not due to refractoriness referred to as ISIs. As observed in neocortical neurons in vivo, we observed action potentials with irregular ISI's that followed a Poisson-like process. From a set of 224 neurons analyzed from four different organoids, 16% ± 8% of the total units fit a Poisson distribution (Fig. 3) with, by definition, the CV approaches one for a perfectly homogenous Poisson process, whereas purely periodic distributions have CV values of zero. Thus, a minority fraction of ISIs were highly irregular (Fig. 3), whereas a majority displayed comparatively more regular spiking patterns with less variation (denoted by a lower CV), which may function to send lower-noise spike-rate signals. ISI

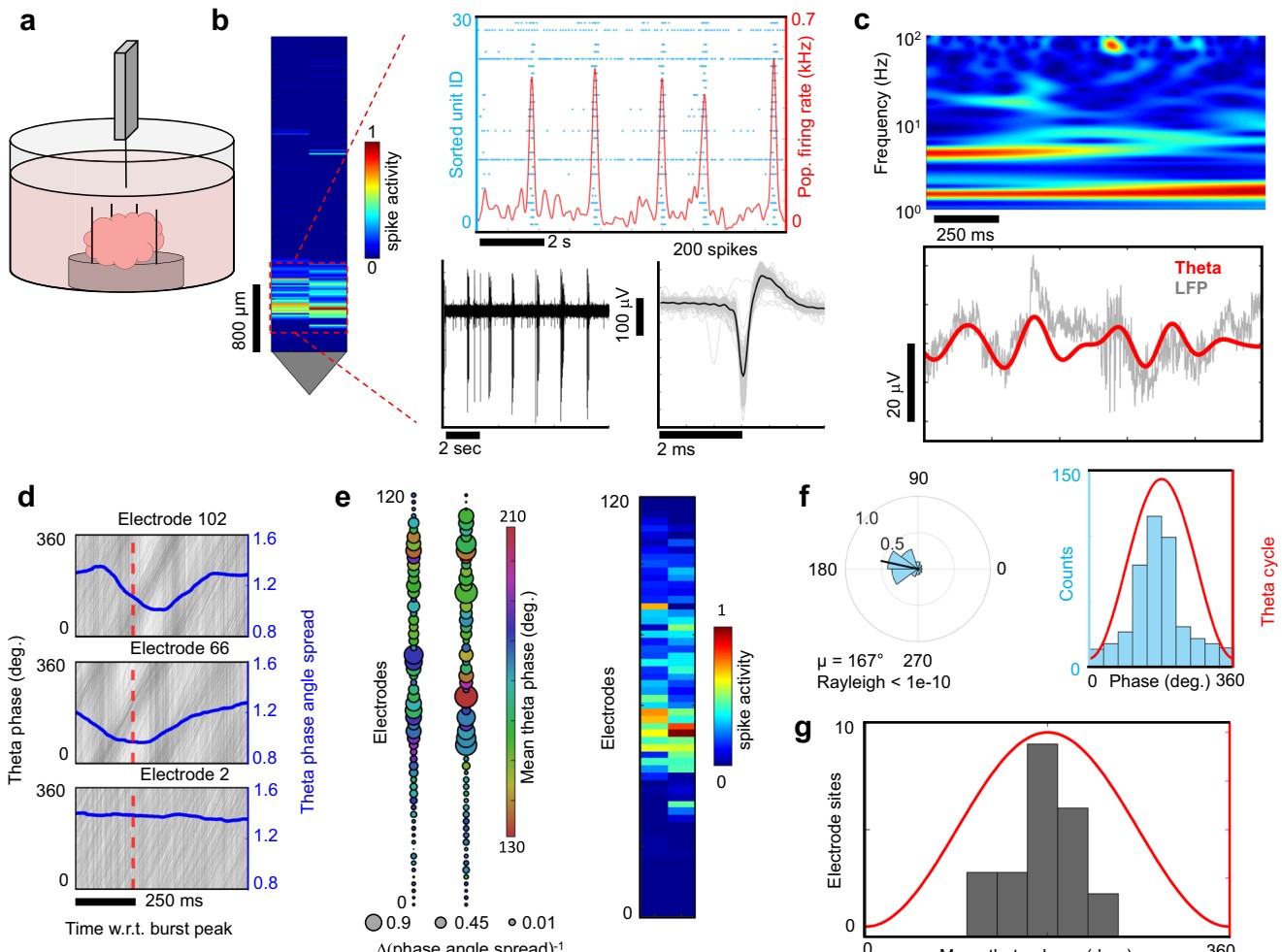

**Fig. 9 | Neuropixels CMOS shanks resolves spiking and LFP in the z-plane of a whole brain organoid. a** A Neuropixels high-density CMOS shank was attached to a custom-made mount and controlled by a micromanipulator in order to lower the shank into an immobilized brain organoid kept at 37 °C in BrainPhys media. **b** Left, spikes (above a 5-rms threshold) were detected in a subset of electrodes near the tip of the shank. Spike activity is normalized relative to the electrode with the most detected spikes. Top right, raster-plot visualization of single-unit spiking activity (Kilosort2) as shown by the blue dots for each sorted unit. Stereotyped population bursts are visualized as peaks in the population averaged firing rate (red line). Bottom right shows the extracellular field potentials (0.3–4 kHz) generated by a spiking neuron within the organoid as measured by the shank. **c** Top, spectrogram plot of local field potential (LFP) from an electrode illustrates dominant oscillation power in the theta and delta bands. Bottom, raw LFP (gray line) from the same electrode overlaid with the theta-filtered band (red line). **d** Theta phase traces (gray lines) are shown relative to population burst peak events (red dotted line) reveal

phase coherence as illustrated by a drop in the phase angle spread plotted in radians (blue line). The bottom plot shows an electrode site displaying no phase coherence relative to population burst events. **e** Left, spatial map of the change in theta phase angle spread is shown relative to burst peak time across the shank. The bubble size indicates the inverse of the phase angle spread relative to the burst peak for each electrode. Notice the overlap with phase coherent sites and the spatial region registering spiking activity on the right. **f** Left, circular distribution of theta-spike phase angles measured from a single electrode site. The direction of the mean spike angle (μ) relative to the theta phase and magnitude (mean resultant length) are shown as polar plots. The Rayleigh criteria for non-uniformity was used to determine if spikes were distributed non-uniformly over the theta cycle (0°, 360°). Right, distributions of theta-spike angles shown relative to the theta cycle. **g** Mean theta phase angle of spike phase-locked electrodes across the Neuropixels probe satisfying the Rayleigh test for non-uniformity ($p < 0.05$).

distributions have also been fitted to gamma distributions that are mathematically equivalent to an exponential distribution when the shape parameter ($k$) is one and converges to a normal distribution for large $k$, thus providing a useful measure of ISI-regularity similar to the CV[28]. Depending on architectonically defined brain regions with specialized cellular compositions and intrinsic circuitry, neurons process information differently[67–69]. Indeed, neuronal firing varies considerably across cortical regions of monkeys[28,70,71]. Therefore, different organizational features across the brain organoid may exhibit different dynamics to account for the observed ISI distributions. The minority fraction of irregular ISI distributions may be a feature of higher levels of entropy and circuit complexity and contain increased capacity for computation and information transfer as found in prefrontal cortex compared to more regular firing patters found in motor regions[28].

We derived a graph of weighted edges that couple single unit node pairs to send and receive spikes over a wide spatial range. Due to the thickness of our organoid slices, many neurons in the slice are too far from any electrode for their spikes to be detected[53]. Thus, we cannot rule out the possibility that intermediate undetected neurons may account for the coupling between two correlated units. The graph does not imply downstream or upstream routes of information transfer beyond the individual binary couplings. Importantly, what the network does demonstrate is a non-random pattern of a relatively small number of statistically strong (reliable) couplings against a backdrop of weaker couplings. As demonstrated in the murine brain[51,52], high anatomical connection strength edges shape a non-random framework against a background of weaker ones (Fig. 6 and Supplementary Fig. 14). The majority of the singe units (nodes), which

we refer to as brokers, have large proportions of incoming and out-going edges. The dynamic balance among receivers and senders could likely reflect short-term plasticity[72].

Brain organoids—composed of roughly one million cells—have neuronal assemblies of sufficient size, cellular orientation, connectivity and co-activation capable of generating field potentials in the extracellular space from their collective transmembrane currents. The basis for low frequency LFPs may be the cellular diversity that emerges in the organoid from the variety of GABAergic cells (Fig. 2), consistent with their role in the generation of highly correlated activity networks detected as LFPs[31], parvalbumin cells (Fig. 2c), associated with sustaining network dynamics[73], and axon tracts that extended over millimeters (Fig. 2b). Coherence of theta oscillations over spatial extents of the organoid was observed and was unlikely due to volume conduction from distant sources, as happens in EEG and MEG measurements[54], because the voltage recordings were conducted within a small tissue volume ($\approx$3.5 mm$^3$). Consistent with minimal volume conduction effects, we validated theta oscillations by demonstrating that the imaginary part of coherency[54] projected onto the same spatial locations identified by cross-correlation analysis (Supplementary Fig. 19). Correlations between theta oscillations and local neuronal firing (Fig. 7) strongly supported a local source for the rhythmic activity[19,20,53]. The local volume through which theta dispersed extended to the z-dimension as shown with the Neuropixels shank (Fig. 9).

In an in vivo setting, theta oscillations have been proposed to define temporal windows for local circuit interactions. These circuit oscillations can subdivide independent computational tasks[60] by encoding and consolidating information within temporally distinct oscillation phases. Unique phase relationships emerge within independent populations of interneurons and pyramidal cells that are activated in a brain-state-dependent manor[62]. Brain organoids represent a system in which excitatory and inhibitory circuits, including parvalbumin-positive interneurons which are crucial for circuit timing[33], spontaneously self-organize in a fashion that reflects aspects of phase-locking (Fig. 8) previously relegated to complex circuits found exclusively in vivo. Our observations shed new light on the potential utility of brain organoid systems to study complex interactions within a diverse network of cell types grown under laboratory conditions from human iPSCs. Furthermore, basic features of network activity generated by organoid slice cultures were also recapitulated in acute measurements performed in whole brain organoids (Fig. 9). These encompass the generation of neuronal population bursts, spatially correlated theta oscillations and a sub-population of neurons phase-locked to those oscillations. Our findings suggest that organoid slice preparations contain sufficient preservation of the anatomical organization necessary to reproduce many of the physiologic metrics found in in vivo systems. In a similar vein, previous work has demonstrated that spontaneous activity generated by cortical slice cultures[74] recapitulates neuronal activation patterns and synchronization with LFPs found during spontaneous activity observed in the cortex of awake monkeys[56]. In fact, long-term organoid slice cultures are proving to be an attractive model system capable of long-term neuronal survival, anatomical reproducibility[12] and displaying long range axonal projections crucial for modeling feed-forward and feed-back circuits germane to CNS development[11].

Given the complete decoupling of the brain organoid from outside input and motor output upon which the brain constructs a closed loop system, the extent of organized ab initio activity in the brain organoid is surprising. Features such as local ISI distributions, non-random network formation, and subpopulations spikes phase-locked to theta oscillations all arose spontaneously in the absence of any task. A neuronal system in which activity occurs in the absence of sensation fits what O'Keefe and Nadel reference as a physiologic scaffold capable of representing information from the external environment before it is experienced[75]. Buzsáki makes a similar point[76] "...the brain already starts out as a nonsensical dictionary. It comes with evolutionarily preserved, preconfigured internal syntactical rules that can generate a huge repertoire of neuronal patterns." The seeds of this concept existed long before organoids or even before neuroscience in the thinking of Immanuel Kant who described synthetic a priori knowledge as independent of experience, but hardly suspected a priori knowledge consisted of nothing more than abstract waveforms.

## Materials and methods
### Brain organoid generation
Human iPSCs were donated by Karch[77]. Briefly, iPSCs were reprogrammed from skin fibroblasts obtained from skin biopsies collected following written informed consent from the donor and approved by the Washington University School of Medicine Institutional Review Board and Ethics Committee (IRB 201104178, 201306108). The control iPCS line F12442.4 was cultured in mTeSR1 medium (Stem Cell Technologies) on tissue culture plates coated with hESC-qualified Matrigel (Corning). mTeSR1 was exchanged every other day and iPSCs were routinely passed using ReLeSR (Stem Cell Technologies). Brain organoids were generated after the method by Lancaster et al.[8] with minor modifications. iPSCs were incubated in 0.5 mM EDTA in D-PBS for 3 min before dissociation in Accutase for 3 min at 37 °C. After adding mTeSR1 and triturating to achieve a single-cell suspension, cells were pelleted for 3 min at 1000 rpm. Cells were resuspended in low bFGF hES media (20% KOSR, 3% ES-FBS, 1x GlutaMAX, 1x MEM-NEAA, 1x β-Mercaptoethanol, 4 ng/ml bFGF in DMEM/F12) supplemented with ROCK inhibitor (50 μM) and plated in U-bottom ultra-low attachment plates at 4500 cells per well. On day 2, low bFGS hES media with ROCK inhibitor was replaced. Another media change was performed on day 4, with omission of bFGF and Rock inhibitor. On day 5, embryoid bodies were transferred to neural induction media (1x N2 supplement, 1x GlutaMAX, 1x MEM-NEAA, 1 μg/ml Heparin in DMEM/F12), and media was replaced on days 7 and 9. On day 10, each neuroepithelial structure was embedded in 15 μl hESC-qualified Matrigel followed by incubation in neural induction media for another 2 days. On day 12, neural induction media was replaced with NeuroDMEM-A media (0.5x N2 supplement, 1x B27 supplement without Vitamin A, 1x β-Mercaptoethanol, 1x GlutaMAX, 0.5x MEM-NEAA, 250 μl/l insulin solution, 1x Pen/Strep in 50% DMEM/F12 and 50% Neurobasal). From day 19 on, organoids were cultured in NeuroDMEM + A media (0.5x N2 supplement, 1x B27 supplement with Vitamin A, 1x β-Mercaptoethanol, 1x GlutaMAX, 0.5x MEM-NEAA, 250 μl/l insulin solution, 12.5 mM HEPES, 0.4 mM Vitamin C, 1x Pen/Strep in 50% DMEM/F12 and 50% Neurobasal) with media changes twice per week. From day 21 on, organoids were kept on an orbital shaker at 75 rpm. Mycoplasma testing was routinely performed on organoids and iPSCs using the MycoAlert Mycoplasma DetectionKit (Lonza).

### Immunohistochemistry and microscopy
Organoid samples were fixed in 4% paraformaldehyde in 0.1 M sodium cacodylate buffer (pH: 7.4) overnight at 4 °C. After rinsing, samples were embedded 10% agarose and sectioned (100 μm) using a vibratome (Leica, Lumberton, NJ). Primary antibodies used were anti-Parvalbumin (1:200; abcam, Cambridge, MA; ab11427), anti-GAD65 (1:200; GeneTex, Irvine, CA; GTX113192), anti-SMI312 (1:500; BioLegend, San Diego, CA; 801701), anti-GFAP (1:500; abcam, Cambridge, MA; ab53554), anti-connexin 43 (1:200; Millipore, Burlington, MA; MAB3067), anti-MAP2 (1:500; GeneTex, Irvine, CA; GTX82661), anti-synaptobrevin (1:500; Synaptic Systems, Goettingen, Germany; 104-211). Samples were viewed through a ×20 oil immersion lens (N.A. 0.85) and imaged using an upright Olympus Fluoview 1000 laser scanning confocal microscope (Center Valley, PA) equipped with a motorized stage utilizing Olympus Fluoview software version 4.2. High-resolution wide-field mosaics were produced as described elsewhere[78].

### Single-cell RNA sequencing of brain organoids

Individual brain organoids were subjected to dissociation using the Worthington Papain Dissociation System (Worthington). Solutions were prepared according to the manufacturer's instructions. Organoids in NeuroDMEM + A (defined in Brain organoid generation) media were cut into pieces with a sterile scalpel blade and washed twice in PBS prior to dissociation. Incubation in papain/DNaseI solution was performed at 37 °C and 5% $CO_2$ with occasional agitation. After 30 min of incubation, organoids were triturated ten times with a fire-polished glass Pasteur pipette, followed by further incubation and trituration every 15 min until dissociation was complete. The total incubation time in papain/DNaseI solution was 90 min. After adding Inhibitor Solution and DNaseI in EBSS, the dissociated cells were pelleted by centrifugation for 3 min at 300 g. The cells were resuspended in 500 μl ice-cold PBS/0.01% BSA using a wide-bore plastic pipette tip and passed through a 35 μm cell strainer. An aliquot of the single-cell suspension was mixed with Trypan blue and analyzed for viability and cell concentration on a Countess II FL Automated Cell Counter (Thermo Fisher). Viability was >90% for all organoid samples.

### Drop-seq library preparation

Following dissociation, cells were diluted to 110 cells/μl in PBS/0.01% BSA. Drop-seq was performed as described in Macosko et al.[79]. Briefly cells and barcoded beads (Chemgenes, 132 beads/μl in lysis buffer) were run on an aquapel-treated microfluidic drop-seq device (Flow-JEM) for co-encapsulation in nanoliter-sized droplets. After droplet-breakage, reverse transcription and Exonuclease I treatment, beads were counted, and 2500 beads were apportioned per PCR tube for cDNA amplification. The amplified cDNA libraries were purified using SPRI beads (Beckman Coulter) and quantified on a Fragment Analyzer (Agilent). Tagmentations were performed using the Illumina Nextera XT kit (Illumina), and the resulting libraries were purified in two consecutive rounds of SPRI beads-based size selection (0.6x beads to sample ratio followed by 1x beads to sample ratio). The size and concentration of the final libraries were measured on a Fragment Analyzer and a Qubit Fluorometer (Thermo Fisher), respectively. Libraries were sequenced on an Illumina Nextseq500 instrument.

### Drop-seq data analysis

Counts matrices were generated using the Drop-seq tools package (version 1.13)[79]. Briefly, raw reads were converted to BAM files, cell barcodes and UMIs were extracted, and low-quality reads were removed. Adapter sequences and polyA tails were trimmed, and reads were converted to Fastq for STAR alignment (STAR version 2.6). Mapping to the human genome (hg19 build) was performed with default settings. Reads mapped to exons were kept and tagged with gene names, beads synthesis errors were corrected, and a digital gene expression matrix was extracted from the aligned library. We extracted data from twice as many cell barcodes as the number of cells targeted (NUM_CORE_BARCODES = 2x # targeted cells). Downstream analysis was performed using Seurat 3.0[80,81] in R version 3.6.3. An individual Seurat object was generated for each sample, and objects were merged using the Seurat merge function. Cells with <300 genes detected were filtered out, as were cells with >10% mitochondrial gene content. Counts data were log-normalized using the default NormalizeData function and the default scale of 1e4. The top 3000 variable genes were identified using the Seurat *FindVariableFeatures* function (*election.method* = "vst", *nfeatures* = 3000), followed by scaling and centering using the default *ScaleData* function. Principal Components Analysis was carried out on the scaled expression values of the 3000 top variable genes, and the cells were clustered using the first 30 principal components (PCs) as input in the FindNeighbors function, and a resolution of 0.4 in the *FindClusters* function. Non-linear dimensionality reduction was performed by running UMAP on the first 30 PCs. Following clustering and dimensionality reduction, putative cell

doublets were identified using DoubletFinder 2.0.3[82], assuming a doublet formation rate of 5% and calculating homotypic proportions based on clustering at resolution 0.4 as described above. A pk value of 0.61 was selected for doublet identification based on the results of *paramSweep*_vs, *summarizeSweep* and *find.pK* functions of the *DoubletFinder* package. After filtering out the doublets as identified by *DoubletFinder*, counts data were extracted, and a new Seurat object was generated. Normalization, variable gene selection, scaling, clustering and UMAP dimensionality reduction were performed as described above with the exception that 32 PCs were used as input for clustering and UMAP. Clustering was performed at different resolutions (0.3, 0.4, 0.6 and 0.8). Clustering at resolution 0.4 resulted in 9 clusters and was in good agreement with the expression of known marker genes for cell types found in brain organoids (Supplementary Fig. 5b). This resolution was therefore selected for delineating different cell populations. Marker genes were identified for each cluster using the Wilcoxon rank sum test implemented the FindMarkers function, with a logFC threshold of 0.25 and Bonferroni correction for multiple testing, confirming and refining our initial cell type annotation using canonical marker genes.

### Interfacing brain organoid sections with high-density CMOS microelectrode arrays

Whole brain organoids were allowed to mature to 4–6 months (see Brain organoid generation section) and were then embedded in 10% (w/v) low melting point agarose at ≈40 °C and allowed to cool at 2 °C for ≈10 min. Organoids embedded in the agarose gel were then sectioned into 500 μm thick slices with a vibratome (Leica VT1000S) using an advance speed of one and a vibration speed of nine. The gel block containing the organoid was mounted onto the surface of the vibratome cutting block with super glue and placed into the cutting reservoir filled with ice-cold cutting solution composed of 50% (v/v) DMEM (Thermo Fisher), 50% (v/v) Neurobasal (Thermo Fisher), 1x penicillin streptomycin (Thermo Fisher). The space surrounding the vibratome liquid reservoir was filled with ice to keep the cutting solution temperature constant during sectioning. The vibratome blade was allowed to advance through the front-face of the gel block and organoid tissue, while leaving ≈3 mm gel uncut and attached at the back of the gel block. This method allowed previous sections to remain attached to the gel block, which minimized torque on the organoid during subsequent sections and prevented deforming or releasing of the remaining uncut tissue embedded in the gel below. Next, the contiguous layer stack of 500 μm thick organoid sections remaining in the gel block (Supplementary Fig. 1a) was then peeled off using sterile forceps (in a sterile dissection hood equipped with a stereoscopic microscope) and placed into individual wells in a 6 well plate filled with cutting solution using a cut p1000 pipette tip. Organoids in the range of 4–6 months of age have a typical diameter of around 3 mm and usually ≈ six 500 μm thick slices cab be obtained per organoid (Supplementary Fig 1a, b). Slices in the upper and lower third of the organoid yield the optimal surface area to volume ratios of viable organoid tissue for interfacing with MEAs for electrical recording. Slices at the top and bottom are of the organoid are generally small and around ≤1 mm in diameter, whereas slices in the center are the largest, but can have a necrotic core and were avoided. The slices were then transferred to new set of wells (6 well plate) filled with cutting solution before transferring to individual wells in a 12 well plate filled with BrainPhys neuronal medium (StemCell Technologies) with the addition of the following supplements 2% (v/v) NeuroCult™ SM1 Neuronal Supplement (StemCell Technologies), 1% (v/v) N2 Supplement-A (StemCell Technologies), 20 ng/ml Recombinant Human Brain-Derived Neurotrophic Factor (StemCell Technologies), 20 ng/ml Recombinant Human Glial-Derived Neurotrophic Factor (StemCell Technologies), 1 mM Dibutyryl-cAMP (PeproTech), 200 nM ascorbic acid (StemCell Technologies), 1x penicillin streptomycin (Thermo

Fisher) and allowed to recover for 24 h in a 5% $CO_2$ incubator at 37 °C prior to placement and positioning on the CMOS MEAs.

High-density CMOS MEAs and custom machined (UCSB physics machine shop) liquid reservoir lids made from Delrin (McMaster-Carr), fitted with $CO_2$ permeable, water vapor impermeable membranes[83], were first sterilized in a 70% (v/v) ethanol solution in deionized water (18.2 MΩ·cm) for 30 min. The liquid reservoir of the CMOS MEA was then rinsed thoroughly with sterile ultra-pure distilled water in a sterile hood (≈5 ml per CMOS MEA well), and the lids were allowed to air dry in the sterile hood. The recording surfaces of the MEAs were then coated by adding 0.5 ml of poly-l-lysine (PLL) (Sigma-Aldrich) solution (0.1 mg/ml in ultra-pure water) into the CMOS MEA reservoirs and then placed in an incubator for ≈1 h at 37 °C with the lids on. The arrays were then transferred to a sterile hood, the PLL solution was aspirated off and washed an additional 3x with ultra-pure water (1 ml volume for each wash). The CMOS MEA reservoir was then filled with 0.5 ml of BrainPhys medium and the lid was placed on it. The CMOS MEA was transferred to a sterile dissection hood equipped with a stereoscopic microscope. Organoid sections were transferred to the CMOS MEA well with a cut P1000 pipette tip and gently positioned over the recording electrode surface of the CMOS MEA with sterile forceps while visualizing with the stereoscopic microscope (Supplementary Fig. 1b). A sterile custom harp slice grid was used to seat the organoid slice to the CMOS MEA surface. The harp slice grid was made by attaching 122 µm nylon fibers (taken from plastic mesh 9318T45, McMaster-Carr) spaced at a 1 mm pitch to a stainless-steel washer (M3 lock washer, McMaster-Carr) with epoxy (Devcon). The organoid sections on top of the CMOS MEAs were maintained in an incubator (5% $CO_2$ at 37 °C), and media was exchanged twice a week.

### Electrophysiology recordings
High-density extracellular field potential recordings were measured in a cell culture incubator (5% $CO_2$ at 37 °C) using CMOS microelectrode array technology (MaxOne, Maxwell Biosystems, Zurich, Switzerland) containing 26,400 recording electrodes with a diameter of 7.5 µm at a center-to-center distance of 17.5 µm within a sensing area of 3.85 mm × 2.1 mm. A subset of which 1024 electrodes could be selected for simultaneous recording[24]. The low-noise amplifiers have a high-pass filter (0.5 Hz) to minimize drift. Electrical measurements were performed once a week (24 h after a media change), and took around 2 weeks after sectioning to start spiking with a marked increase in activity in the form of synchronized bursts at ≈6 months in age (Supplementary Data 1), reaching a peak in activity at around 7 months (Supplementary Fig. 2a). Automatic activity scans (tiled blocks of 1020 electrodes) were performed to identify the spatial distribution of electrical activity across the surface of the organoid. Data were sampled at 20 kHz for all recordings and saved as HDF5 file format. We chose the routed top most spiking 1020 electrodes to have a minimum spacing distance of at least two electrodes (2 × 17.5 µm), providing sufficient electrode redundancy per neuron to enabling accurate identification of singe-units by spike sorting[26], while simultaneously resolving network activity across the organoid (Fig. 1).

### Spike sorting
The raw extracellular recordings were automatically spike sorted to extract single-unit activity. The same spike sorting procedure was applied to the planar HD-MEA and the Neuropixels recordings. The raw traces were band-pass filtered between 300–6000 Hz before applying the Kilosort2 algorithm[25]. The spike sorting output was automatically curated by removing units with an ISI violation threshold[84] above 0.3, an average firing rate below 0.05 Hz, and a signal to noise ratio below five. For processing data from the same chip with the same electrode configuration, recorded within a short period of time (see stability section in "Methods"), we concatenated the filtered traces in time

before applying spike sorting. All of the processing was performed using the SpikeInterface framework[26].

### MEA local field potential signal processing
Raw signals (sampled at 20 kHz) were imported into MATLAB (Mathworks). The LFP component was extracted by first low pass filtering the raw data (frequency cutoff of 500 Hz using 4th order Butterworth filter), and downsampled to 1 kHz. The LFP was then filtered into the traditional oscillatory sub-bands using a bi-directional finite impulse response filter[85,86]. Delta (0.5-4 Hz), theta (4–8 Hz), alpha (8–13 Hz), beta (13–30 Hz) and gamma (30–50 Hz) bands were extracted using *eefilt.m* function from the UCSD EEGlab MATLAB toolbox (Supplementary Fig. 15).

### Acute recordings from brain organoids using a high-density Neuropixels CMOS shank
Six-month-old brain organoids were transferred to BrainPhys media and were cultured for 30 days without shaking before recording. Organoids were then positioned using a cut p1000 pipette tip into a custom well (filled with BrainPhys media) to immobilize the organoid (kept at 37 °C on a temperature-controlled stage). A high-density CMOS shank with 960 electrode sites that tile along a 70 × 20 µm cross-section shank (10 mm long) where 384 electrodes can be programmably routed and recorded from simultaneously[27] was inserted into the organoid using a low drift (<10 nm/h), motorized precision micromanipulator (MP-285, Sutter Instruments). The Neuropixels probe was mechanically attached to the micromanipulator using a custom machined bracket (UCSB physics machine shop). Electrophysiology data were acquired using SpikeGLX (Bill Karsh, https://github.com/billkarsh/SpikeGLX). Action potential channel was sampled at 30 kHz. The local field channel (sampled at 2500 Hz) was low pass filtered at 500 Hz. Downstream processing and analysis were performed with methods similar to 2D CMOS data.

### Single-unit interspike interval distributions
The selection and filtering of single-unit activity was based on the work of Maimon and Assad[28]. First, single-units were selected for analysis that had a minimum of 30 spikes measured over a 3-min interval and ISI intervals with a range of 8 ms to 100 ms were only considered for further analysis. An exponential ($f(x) = a \cdot e^{-\lambda x}$) function was fit using a non-linear regression model (*fitnlm* function in MATLAB) to normalized histograms (*histcounts* function in MATLAB) of the single-unit ISI distributions (bin widths were set to 1/15th of the median ISI for a given unit). Here, $f(x)$ represents the normalized bin counts for a given ISI bin designated by $x$. The constants $\lambda$ and $a$ were free fitting parameters. Regression fits were selected that had $R^2$ values >0.9 to classify ISI distributions as exponentially distributed. A two-sample KS test was performed to assess differences between ISI distributions under control and 50 µM diazepam conditions. All statistical tests performed in the manuscript were two-sided unless specified otherwise.

### Population-averaged spike rate and population bursts
For each recording, a population averaged firing rate vector was computed for the spike sorted single-units by summing the spiking activity over all units for each recording frame and averaging these summed values with a 5 ms moving average. For each burst, a burst population rate (5 ms moving average and 5 ms Gaussian) was selected by taking a window with respect to the peak of the MUA (defined below). For each pair of bursts, the similarity in the population rates were assessed by taking the average of the absolute difference between the two burst population rate vectors. This difference was then averaged over all burst pairs. Burst-to-burst similarity between control and diazepam was compared by computing the ratio in the average population rate vector difference between control and diazepam $(\rho_C - \rho_D)/(\rho_C + \rho_D)$, where $\rho_C$ and $\rho_C$ are the population rate

differences for control and diazepam, respectively. This analysis was performed using a range of time windows with respect to the peak of the MUA activity. Window start times were selected between −200 ms until 0 ms with respect to the MUA peak with a step size of 10 ms. Window end times were selected between 0 ms until 500 ms with respect to the MUA peak with a step size of 10 ms. The average ratio over all the different time windows was computed to compare the control and diazepam treatments for the different organoids.

MUA was extracted by band-pass filtering (cutoff frequencies of 0.3 and 7 kHz using a 4th order Butterworth filter) the raw voltage signal, and spike detection was set at 5x the rms-noise for a given electrode[87]. Detected spike times (rounded to the nearest ms) were then stored as a binary spike time matrix (columns represent electrodes and rows represent time). MUA population-averaged spike rates were calculated by first summing all detected spikes (across all 1020 electrodes) occurring within a one ms time window to create a "population spike" vector. The "population spike" vector was then averaged over a 20 ms sliding window and smoothed with a 100 ms Gaussian kernel, yielding a population-averaged spike rate as measured across the CMOS array. Next, population bursts peaks were defined when the population-averaged spike rate exceeded 2x its rms value (using the *findpeaks* function with a minimum peak distance of 1 s). Population burst durations were defined by time windows surrounding the burst peak where the burst amplitude was attenuated by 90% of its maximum value.

## Spike time tiling coefficient to draw a functional connectivity map

Correlated spiking activity between single-unit spike trains was compared using the STTC[49]. A publicly available MATLAB script[11] was used to compute the STTC. A correlation window ($\Delta t$) of 20 ms was used to compute a score of functional connectivity strengths across all units. The remaining correlations between single unit pairs were further filtered based on the spike time latency distributions of the spike trains. All electrode pairs with a multimodal latency distribution (Hartigans' dip test $p < 0.1$), a full width at half maximum larger than 15 ms were removed. To rule out the influence of outliers, these measures were computed over all spike time latencies within a [−20;20] ms range. Using the STTC scores of the remaining unit pairs, functional connectivity networks were generated.

## Lower bound connectivity score estimation

To obtain a lower bound for connectivity scores to use for generating connectivity networks, STTC connectivity scores for randomized spike time vectors were computed. Randomization was performed by selecting the spike times within an individual burst event and shuffling the corresponding electrode numbers. This processing was done for each burst event individually and resulted in randomized spike time vectors while maintaining the same population rate as the original recordings. Connectivity scores between electrode pairs were computed in the same way as the original data. Based on the distributions of the STTC scores of the remaining connections, a lower bound threshold of STTC = 0.35 was chosen. Within the randomized data sets ($n = 6$ organoids), 93.2% ± 14.7% (mean ± STD) percent of the STTC connectivity edges strengths were below the 0.35 threshold across 13 separate measurements; organoid L6 had no pairwise spike correlations (edges) after spike train randomization and was excluded from the statistics above (Supplementary Fig. 11b for randomization distributions).

## Global connectivity network metrics

Global and local connectivity metrics for the connectivity networks were analyzed using the MIT strategic engineering network analysis toolbox for MATLAB[88]. For networks generated using an STTC, all edges were removed that fell below a threshold of 0.35 (see "Lower bound connectivity score estimation" detailed above). The size of the largest and second largest component, defined as the number of nodes present in the largest and second largest structure of connected nodes. The size of the second largest component was <2% of the size of the largest component for L1, L2, L3 and L4 and <10% for L5 and L6 for 7-month-old organoids.

## Directional connectivity map

Based on the mean of the spike time latency distributions, directionality was assigned to the edges in the functional connectivity map. A negative mean spike time latency compared to the reference unit indicated a directionality toward the reference unit and vice versa. Based on the directionality, the in-degree ($D_{in}$) and out-degree ($D_{out}$) were computed per unit, defined as the number of incoming and outgoing edges respectively. "Receiver" nodes were defined as $(D_{in} - D_{out})/(D_{in} + D_{out}) > 0.8$, while "sender" nodes were defined as $(D_{out} - D_{in})/(D_{in} + D_{out}) > 0.8$. All other nodes were labeled "brokers". The directionality per node was calculated for the largest component of interconnected nodes. To evaluate the significance of "sender" and "receiver" nodes, networks were randomized by performing double edge swaps (ten times as many swaps were performed as edges in the control network) and contained the same number of nodes and the same edge density. Randomized networks contained no "sender" or "receiver" nodes.

## Diazepam treatment and analysis

Diazepam (Sigma) was solubilized in DMSO (Sigma) at 1000 times the target concentration and 1 μl was added to the 1 ml CMOS MEA reservoir containing culture media and gently mixed with a p100 pipette. The cultures ($n = 4$) were then allowed to sit undisturbed for 15 min in the incubator (37 °C 5% $CO_2$) before recording. Organoid dose responses were carried out over 3, 10, 30 and 50 μM diazepam concentrations (Supplementary Fig. 7).

For connectivity networks generated using recordings from organoids, treated with 50 μM diazepam, the shape of the edge strength distributions in the largest component then were compared. A power law fit ($y = a \cdot x^b$), an exponential fit ($y = a \cdot e^{-c \cdot x} + d$) a truncated power law fit ($y = a \cdot x^b \cdot e^{-c \cdot x}$) and a Gamma distribution fit ($y = (a/(\Gamma(b) \cdot c^b)) \cdot x^{b-1} \cdot e^{-x/c}$ were performed on the binned edge strength (pairwise spike train correlations determined by the spike time tile coefficient) distribution (bin size = 0.05). The goodness of the fit was assessed using the Akaike information criterion (AIC)[50,89] (Supplementary Fig. 14e). Organoid L4 was excluded due low number of edges relative to the other samples.

## Edge change

A gamma distribution was found to be the optimal fit (lowest AIC score, Supplementary Fig. 14e) and used to further assess differences in the distribution of pairwise spike correlation strength determined by the STTC. The fractional change correlation strengths between pairwise spike trains was defined as: $(f_d - f_c)/(f_d + f_c)$, where $f_c$ and $f_d$ are the fraction of total edge strength distribution fits to the gamma distribution for control and diazepam (50 μM), respectively. Pairwise spike correlation distributions were split into three unique sets of edges. The first set contained edges that were present in both the control and the diazepam recording. The second set contained edges that were silenced by diazepam, meaning that they were only present under control conditions. The third set contained edges that were induced by diazepam, meaning that they were only present under the diazepam condition. Pairwise spike train correlation distributions and connectivity maps were generated for the three different sets of edges.

## Time course in culture analysis

Electrical recordings were made from organoids at 6, 7 and 8 months were compared from three organoids. First, the MUA population-

averaged firing rate (defined above) was calculated for each burst period. The results were averaged over all bursts within a recording centered about the population burst peak. Secondly, the peak MUA population rate was averaged over all bursts within a recording and compared between organoids at different ages. The average inter-burst interval, defined as the average time between burst peaks was compared. Functional connectivity maps based on the STTC were generated for the organoids at the different time points.

## Stability analysis

Electrical recordings were made from organoids spaced 4 h apart. The number of spike sorted singe-units detected in each recording were compared for two separate organoids (Supplementary Fig. 13a). In addition, pairwise spike correlation scores were computed between all single-unit pairs. The pairwise spike train correlation distributions for the two measurement time points were compared by a two-sample KS test (Supplementary Fig. 13b). Separate measurements taken from 3-min long recordings were split in half (first and second 90 s intervals), pairwise spike correlations were calculated for the two time intervals and their distributions were compared using two-sample KS test for three organoids (Supplementary Fig. 13c).

## Theta coherence

Normalized cross-correlation analysis was performed between all pairwise electrodes (1020) for the theta band time series using the *xcorr* function in MATLAB. The correlation coefficient and respective lag-time were stored in 1020 × 1020 matrices. Before performing pairwise correlations, theta oscillation amplitudes smaller than 10 μV were excluded and considered below the noise floor of the CMOS detectors at this bandwidth (Supplementary Fig. 16 for noise floor measurements with synaptic blockers and TTX). Next, each electrode correlation strength was determined by ranking by the summed correlation coefficient between all pairwise electrode sites. The top correlated electrode ("seed" electrode) was then used to generate a spatial correlation and phase-lag map between the other 1019 electrodes across the array. This analysis was also repeated for theta time series occurring during population burst periods (see population-averaged firing rate and population bursts in "Methods" for details). Next, the total number of pairwise theta correlations occurring within the burst periods and during the whole time series were compared. Finally, the total number of pairwise correlations with a correlation score of at least 0.2 was compared between control and diazepam conditions during burst time periods.

To further validate the cross-correlation calculations, theta oscillations were signal averaged relative to the "seed" electrode. First, positive peaks were identified using the *findpeaks* function with a minimum peak height of 10 μV (see Supplementary Fig. 16e and Blockers and TTX analysis in Methods) and a minimum peak distance of 100 ms. Next, the theta peak time points from the "seed" electrode were used as anchor points to average theta signals centered over a 500 ms time window for each electrode. The amplitude and relative phase shifts (relative to the "seed" electrode) were used to generate spatial coherence maps of theta oscillations across the organoid.

## Burst peak-centered theta signal averages

Event triggered averages were made of the theta filtered LFP from each electrode using the peak of the population burst as the anchor point. For each electrode the theta filtered signal was averaged over all burst events. Spatial maps were generated showing the theta phase per electrode at different time points relative to the population burst peak. The amplitude of the envelope of the averaged theta signal was used as a measure for the consistency of the theta phase relative to the MUA population burst peak for multiple burst events. Similarly, the MUA population burst peak-centered theta phase was obtained for each burst of spiking activity (over a 5 ms sliding window). Theta phase angle spread was determined by calculating the circular standard deviation of the theta phase relative to the population burst peaks[90]. The circular standard deviation (defined as the phase angle spread in Fig. 7g) over all bursts of spiking activity was computed for each time point relative to the burst peak. For each electrode where the circular standard deviation fell below one within 250 ms before the burst peak until 500 ms after the burst peak, the time point relative to the peak with the lowest circular standard deviation was obtained. Using these relative time points of lowest circular standard deviation, a histogram showed the time relative to the burst peak with the most consistency in the theta phase of the signals taken over all individual spiking activity bursts.

To visualize spatiotemporal changes in the theta amplitude during the onset of a population burst, a spatial window of ≈140 μm × 140 μm (corresponding to 5 × 5 electrode grid routing with a 35 μm center-to-center spacing) was selected to calculate spatial averages centered around each sampled recording electrode. The theta signals were then averaged over all the electrodes within this spatial window sampled over the duration of the recording. If no electrodes were sampled within the spatial window, the centered electrode was discarded. Next, spatially averaged theta filtered signals were averaged over all burst events relative to the peak in population activity (as described above). Spatial plots were made of the averaged theta signal at different time points with respect to the burst peak to visualize the spatial evolution of theta amplitudes during population bursts.

## Theta activity within and outside of neuronal population bursts

Cross-correlation analysis of the theta band time series, as described above, was performed on the whole time series and as cross correlations on theta time series occurring within population bursts (detailed in "Population-averaged spike rate and population bursts" section of "Methods"). The number of correlation scores of at least 0.2 was compared between whole theta time series data and the theta time series occurring during population bursts. Next, for each electrode that had a correlation score of at least 0.15 with the reference site for the population burst time series, the envelope of the theta filtered LFP was generated and the rms of the envelope was computed. The fraction of the recording for which the envelope exceeded 1.5 times the rms was computed for the recording frames within burst periods and the recording frames outside burst periods. The number of electrodes that showed at least a 10% increase in the fraction of time during which the envelope was above the rms threshold during bursts periods compared to non-burst periods was counted. Similarly, the number of electrodes with a 10% increase during non-burst periods compared to burst periods was counted.

## Imaginary coherence estimation

The raw LFP signal, band-pass filtered into the theta band was used to compute the degree of coherence using the imaginary part of the coherence estimation. The imaginary coherence metric between each of the electrode pairs[54] was computed using a window function with the length of 0.5 s and 25% overlap. The connectivity strength at each electrode, regional imaginary coherence, was estimated by averaging across all Fisher's Z-transformed imaginary coherence values[91]. After generating the imaginary coherence matrix for the full 1020 × 1020 electrode pairs, we thresholded this data-matrix at the 90th percentile of the imaginary coherence values. We selected the nodes with >200 connections in this limited matrix, which resulted in seven highly coherent electrodes (Supplementary Fig. 19b). These seven electrodes were connected to 595 other electrodes in total. These nodes were clustered based on squared Euclidean distance and k-means clustering method (*evalclusters* function in MATLAB). The clustering algorithm with its highest stability, identified two clusters, and separated a small cluster with 54 electrodes, which was locally distributed on the array (cluster 1; Supplementary Fig. 19c) and had tighter within coherence (Wilcoxon ranked sum test, $p < 1e-4$, Supplementary Fig. 19e). The

fraction of electrode sites identified by imaginary coherence calculations and cross-correlation analysis were calculated as a function of correlation threshold to determine the extent of spatial overlap between the two methodologies (Supplementary Fig. 19g, h). In three different organoids, for each of the 1020 electrodes, the average strength of the imaginary coherence was calculated as the row-averages for the 1020 × 1020 matrix, before and after treatment with diazepam. A Wilcoxon ranked sum test was used to compare the pre- and post-diazepam conditions for each of the organoids, demonstrating a significant reduction of imaginary coherence with diazepam treatment (Supplementary Fig. 19f).

### Blocker and TTX experiments and analysis

Preparation of stock solutions to block components of fast synaptic transmission: the AMPA receptor blocker NBQX (Abcam) was solubilized in DMSO, the NMDA receptor blocker R-CPP (Abcam) and the GABA receptor blocker Gabazine (Abcam) were prepared in ultrapure distilled water (Life Tech) at 1000x the desired working concentration. The working concentrations were 10, 20 and 10 µM for NBQX, R-CPP and Gabazine respectively. The sodium channel blocker tetrodotoxin (TTX) citrate (Abcam) was solubilized in ultrapure water at 1000x the working concentration (1 µM). Recordings were made from organoids silenced with synaptic blockers and TTX. Spatial maps of the median firing rate per electrode were generated from organoids treated with synaptic blockers, synaptic blockers in combination with TTX and for the control recordings of untreated organoids. For each treatment, the number of actively spiking electrodes was counted. This was defined as having at least 30 spikes over a 120 s window.

For each electrode on the array, the theta oscillation amplitude envelope was computed[92]. The rms of the theta amplitude envelope was calculated for recordings of the organoids silenced with synaptic blockers and TTX. Subsequently, the fraction of the total recording time that the theta envelope exceeded a threshold of 1, 1.5 or 2 times the rms was computed for each electrode, as well as for the burst periods and the non-burst periods of the same electrode during the control recording. This fraction of time above the threshold during either the burst period or the non-burst period of the control recording was compared to the blockers and TTX recording. The percent change with respect to the blockers and TTX recording was computed and averaged over all electrodes that spiked at least 30 times during the control recording. Similarly, signal envelopes were obtained for the delta, theta, alpha, beta and gamma frequency bands. For each frequency band in each electrode, the rms was computed from the organoids silenced with synaptic blockers and TTX. The fraction of time that the envelope exceeded 2x the rms was computed for the whole time range of the recording treated with blockers and TTX, as well as for the burst periods of the control recording. The percent change of the fraction of time that the envelope exceeded the threshold was computed for each frequency band in each electrode. In addition, the constant decrease in the LFP amplitude over all frequencies after treatment with blockers and TTX was further demonstrated by comparing the power spectral density plots of the same electrode during control recordings and during recordings after treatment with blockers and TTX. The LFP power spectrum scaled inversely with frequency ($-1/f$)[93] and was above the electronic noise floor in control conditions (Supplementary Fig. 16a–d). The mean theta rms voltage fluctuations measured across all electrode for $n = 4$ organoids was $7 \pm 2$ µV (mean ± STD) for blocker + TTX experiments (Supplementary Fig. 16e), which is in agreement with noise thresholds at this bandwidth for this CMOS architecture in physiological environments[24].

### Spike phase locking to theta

The time series theta phase, $\phi(t)$, was determined by taking the standard Hilbert transformation, $H(t)$, of the theta band oscillation and calculating the angle between the real and imaginary components of $H(t)$ in MATLAB[92]. The theta phase was then counted at each spike event time for all spike-sorted units. The Rayleigh criterion was used to test the non-uniformity of spikes distributed across the circular phase angles (0°, 360°)[59,65,90]. Spikes were considered to be phase-locked to theta if they passed the Rayleigh criteria test for non-uniformity ($p < 0.05$).

### Reporting summary

Further information on research design is available in the Nature Research Reporting Summary linked to this article.

## Data availability

The data support the findings of this study are available within the article and its Supplementary Information. Electrophysiology recordings can be found here https://doi.org/10.25349/D9031Z. scRNA-seq data have been deposited and are publicly available in the NCBI Gene Expression Omnibus (GEO; http://www.ncbi.nlm.nih.gov/geo) under accession GSE207749. Source data are provided with this paper.

## Code availability

Spike sorting was performed in Python 3.6 using SpikeInterface 0.13.0 and previously published (Buccino et al.[26]), which can be found here https://github.com/SpikeInterface/spikeinterface. Pairwise spike correlation analysis was performed in MATLAB (version 2018b) utilizing previously published code in C (Cutts and Eglen[49]), which can be found here https://github.com/CCutts/Detecting_pairwise_correlations_in_spike_trains and has been adapted to MATLAB (Giandomenico et al.[11]), which can be found here https://github.com/Timothysit/organoids. Custom code for the visualization of organoid network activity is available at https://github.com/KosikOrganoid/Intrinsic-activity-code.

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

## Acknowledgements

The authors acknowledge the use of the William K. Bowes Laboratory for Stem Cell Biology and Engineering at the University of California, Santa Barbara. Illumina sequencing was carried out at the Biological Nanostructures Laboratory, within the California NanoSystems Institute, supported by the University of California, Santa Barbara and the University of California, Office of the President. This study was funded by the Arnold O. Beckman Postdoctoral Fellowship Award (T.S.), Dr. Miriam and Sheldon G. Adelson Medical Research Foundation (K.S.K.), Larry L. Hillblom Foundation (K.S.K.), National Institutes of Health grant number K08AG058749 (K.G.R.), R01NS100440 (S.S.N.), R01AG062196 (S.S.N.), UCOP-MRP-17-454755 (S.S.N.), Larry L. Hillblom Foundation: 2019-A-013-SUP (K.G.R.), Alzheimer Nederland (T.V.D.M.), Swiss National Science Foundation Early Postdoc Mobility and Postdoc Mobility grants P2ZHP3-174753 and P400PB-186800 (S.M.K.G.), ERC Advanced Grant 694829 "neuroXscales" (A.H.), and the ETH Zurich Postdoctoral Fellowship 19-2 FEL-17 (A.P.B.).

## Author contributions

T.S. and K.S.K. designed, conceived and supervised the study; P.K.H., K.R.T. and L.R.P. offered numerous suggestions and comments; T.S. designed and performed electrophysiology experiments on sliced organoids; T.S., T.V.D.M. and E.G. performed computational analysis and statistics on electrophysiology recordings; E.G. performed shank recording on whole organoids; S.M.K.G. cultured the organoids, performed single-cell RNA sequencing and analysis; A.P.B. performed spike

sorting analysis under the supervision of A.H.; G.L. performed immunohistochemistry, imaging and organoid slicing; K.G.R. and K.K. performed imaginary coherence analysis under the supervision of S.S.N.; Z.C. performed additional analyses.; M.A. contributed to cell culturing; T.S. and K.S.K. wrote the manuscript, and all authors discussed the results and commented on the manuscript.

## Competing interests

K.S.K. is on the SAB of Herophilus. All other authors declare no competing interests.
