## [Peer Review File · Nature Communications]

Functional neuronal circuitry and oscillatory dynamics in human brain organoidsReviewers' Comments:

Reviewer #1:

Remarks to the Author:

The authors provide a very detailed functional map recorded with high-density MEAs from slices of human brain organoids. They demonstrate that these slices, and thereby the organoid itself, are electrically active. Electrical activity is identified at the level of single units, of multi-unit activity and the level of local field potentials. Furthermore, the authors demonstrate that spiking activity is modulated by diazepam, and abolished by TTX, a blocker of (most) sodium channels. As the main message of the paper the authors state that "human brain organoids have self-organized neuronal assemblies of sufficient size, cellular orientation, and functional connectivity to co-activate and generate field potentials from their collective transmembrane currents that phase-lock to spiking activity."

Neither the technology nor the analysis tools are new. The novel aspects from a technological point are the combination of two state-of-the art electrophysiological systems with versatile computational analysis. However, many computational tools had been developed previously. Therefore, my main concern relates to the novelty of the results presented in this manuscript. To the best of my understanding, field potentials have been reported in ref.16 of this manuscript.

The authors state in the second sentence of the discussion: "Over the entire area of the organoid assessed by the MEA this activity has the form of a network in which single-units are considered nodes connected by weighted directionally defined edges". They evaluate "coupling between single-units" from spike train latency distributions (see i.e. Fig. 5b). Using this method they generate a "connectivity map", presented i.e. in Fig.5d, 5f and suppl. Figures. The term connectivity could be misleading here. First of all, it seems as though several cell pairs are connected over an extended distance of up to 4mm, and a spike latency of about 4 ms. This very short time would be necessary to conduct action potentials along the axons. Please comment how "connected" pairs can exist over distances of several millimetres.

An alternative explanation for the observed pattern may be one (or more) so-called central pattern generators. Pattern generator cells may be connected to many cells but not recorded here. Could the authors exclude such scenario?

The authors evaluate theta rhythms and their phase locking to single-unit spiking. The authors need to flesh out this interesting result. Are the theta rhythms stationary or do they move / propagate (see i.e. the study by Ferrea, Maccione et al. 2012 or by Menzler&Zeck, 2011).

Theta rhythms are compared to MUA but not to single-unit spiking. Any comments, why ?

Appropriate blocking experiments (i.e. with low dose of TTX) may inhibit spiking but not LFPs. Could the authors draw a conclusion, if the LFPs persist without (most of the) spiking ?

The presented spike maps are presented for slices of a human brain organoid not the organoid itself. The difference should be made clear already in the header of the subsection.

The authors present data from 6 organoid slices. It would be valuable to mention, how many slices could be recorded per organoid.

Spike phase-locking to theta – The spikes from several electrodes seem to be locked to theta but with various time lags ? What are the implications ?

Reviewer #2:

Remarks to the Author:

The manuscript entitled "Human brain organoid networks" by Sharf and coworkers deals with the recordings of $n = 6$ brain organoids by means of high-density CMOS-based MEAs. The authors exploited the potentiality of the recording systems to evaluate both spiking and LFP activity to extract both dynamical and functional properties of the network. In addition, in $n = 4$ organoids, the authors applied a pharmacological protocol by delivering diazepam to evaluate variations in the recorded

patterns of electrophysiological activity. Finally, $n = 3$ brain organoids have been used for acute recordings by combining the MEA with the Neuropixel shank in order to evaluate electrophysiological activity also in the third dimension.

Indeed, the topic of the manuscript is of great interest for the community of (computational) neuroscientists as well as for the small community of MEA users since it shows the possibility to use such devices to characterize an experimental model that is (without any doubts) more realistic than slices or cell cultures (also in 3D configuration).

However, I found the manuscript not well organized and difficult to follow, starting from the title. Such manuscript seems a collection of nice and interesting results but without an organized pathway.

Here below, I will list some of the main concerns I found after a careful reading.

- The title is too vague and does not transmit any useful message. It completely loses of relevant keywords like "connectivity", "spiking", "LFP", "oscillations", "propagation", "MEA". In addition, the title of the Supplementary Information file is different: "Intrinsic network activity in human brain organoids". Why?

- Pg. 5. Line 94. "'Slicing of organoids...". If I understood well, brain organoids have been sliced and then coupled to MEA in order to enhance the possibility to survive. But, in terms of dynamics, how does change the activity of "an entire organoid"? did the authors perform any attempts?

Consequently, which are the differences with respect to an organotypic slice? The authors should provide evidences that slices, sliced-organoid, and entire organoid display similar/dissimilar patterns of activity. This methodological approach should be also discussed in the Discussion section.

- Figure 1A shows an example of spiking activity map coming from a representative organoid. How do the authors explain that the activity is recorded only at the periphery of the organoid? Did they investigate the coupling in the middle of the slice? Is it possible (and which solutions) to increase the sealing (e.g., by changing the adhesion factors). In my opinion this is a strong limit of the approach, especially (see Figure 5d) if one of the aim is extract the functional connectivity map of (a layer) of the organoid. How many short connections are lost (I am referring to the example of Figure 5d).

- I found a little bit far from the topic of the work, the rigorous characterization of the neuronal composition of the organoid. This is not a novel result. The authors could cite many of the works of the literature that have already detected the different neuronal types. Although nice, is Figure 2 fundamental?

- Since the analyses have been performed with a small sample of organoids, statistics analyses have been often not done. And this is not a problem, and I found much more informative to display the metrics organoid by organoid. What I do not like, that I find not rigorous is the use of the confidence level instead of the p-value in some analysis. For example line 225, pg 10. I would see the p-values.

- At the beginning of the Section "Short-term interactions in human brain organoids", the authors should better explain the meaning of "short-term interactions" (line 296), by introducing (for example) time constants that can also sustain the choice of the used correlation parameters.

- Regarding the correlation analysis, how long are the used recordings? are the achieved results stable? if the authors try to split the recordings to verify the stability of the correlation outcomes? Please, discuss!

- Figure 5c is not fundamental, and can be included as subplot.

- The last statement at pg 14 is weak. I suggest to remove.

- I found very interesting the analysis on the kind of nodes (sender, receiver, broker), but I think that it should be better discussed. Do the authors have an idea of the achieved percentage? How did they choose the threshold 0.8? How robust is such a value?

- The sentence "This result was consistent..." (line 360-362) is weak. I would remove.

- Regarding the "Acute measurements in whole organoids" section, what I think it could be disruptive is the simultaneous recording of the brain slice from MEA (2D) and by the Neuropixel (3D) and then try to extract 3D features.

In 2021, Shin and coworkers published a great work on Nat. Comm where they recorded the activity by means of 3D high-density MEAs and were able to reconstruct the 3D functional connections of the network.

Instead, in this manuscript, the authors only exploited the presence of the shank to evaluate whether

there are similar electrophysiological features of the 2D layer. I know that new experiments are time-consuming, but why the authors do not organize an experimental campaign where CMOS-MEA and Neuropixel are simultaneously used?

- The discussion section should be completely revised. Not all the figures are critically discussed as well as the results are poorly discussed with respect to the state of the art. Which is the most novel thing that should emerge from this manuscript? In the current version of the manuscript, I have difficulties to find it.

- Data and materials availability do not satisfy the current standard. In 2021, it is unacceptable to read "Additional data and code related to this paper are available from the corresponding author upon reasonable request". Please, share data and code with a doi in public repository.

Reviewer #3:

Remarks to the Author:

NCOMMS-21-40091

Sharf et al. have produced a remarkable characterization of neuronal activity in human brain organoids using CMOS-based microelectrode array (MEA) comprised of 26,400 electrodes in the x-y plane and a 960 electrode Neuropixels CMOS shank probe in the z-direction. The conclusions are comparable to those put forth in a recently published article (PMID 34426698), i.e., human brain organoids with excitatory and inhibitory neurons are capable of generating complex oscillatory activities not unlike in the intact brain. Instead of using a disease model, the authors of the present study carried out sophisticated analyses for determining connectivity within the organoid and establishing phase-locked firing of the neurons with theta frequency oscillations. I have a few comments that should be considered for a better documentation of the neuronal activity that takes place in the organoids.

1.- The authors describe the firing of the neurons in the organoids as "bursts". However, looking at the samples of the firing patterns it is evident that the firing is much more complicated than just simple bursts. There seem to be bursts of bursts (clusters) and perhaps even bursts of clusters (superclusters). This nomenclature comes from the single-channel literature, where the patterns of channel openings and closures determine the kinetic schemes assigned to channel activity. I recommend using the log-binned plots of the inter-spike intervals (ISI), as the closed times are plotted for single channels, to determine the nested exponential distributions of the ISIs (e.g., see PMID 6131450). This would be a better graphical representation than the cumulative probability distributions plotted on a linear ordinate (e.g., Fig.3 a & b), where it appears that about 20% of the spikes have 0 Hz firing rate, which is impossible. If the exponential distributions of the ISI hold, then there will be no need for artificially fitting the cumulative Weibull distributions, as these distributions do not seem to fit too well, as illustrated on Fig 3a both for the control and diazepam conditions.

2.- The authors make a good point about the theta frequency oscillations being phase locked to the firing of action potentials. However, on the raw LFP traces there appear to be higher frequency oscillation occurring together with the theta rhythms (e.g., Fig. 7a). Was there any phase-amplitude or frequency-amplitude coupling between low and high frequency oscillations? This would be a nice further demonstration of highly organized network activity being present in the organoids.

3.- It is regrettable that the histology and the recordings are shown on different organoid cross-sections. It would be extremely informative to present the histology even with just a pan-neuronal marker such as NeuN for the electrophysiological map shown in Fig.1a.

4.- This is a minor point, but it is highly unlikely that people with color vision deficiency will get much out from the color scales for the phase in several figures (e.g., Fig. 7f).

5.- The last sentence of the Discussion attributes too much to the activity recorded in the sensory input and motor output deprived organoids. After all, Sherrington stated that "The brain seems a thoroughfare for nerve-action passing its way to the motor animal". Accordingly, without a motor output, the organoids should not even be denoted as "brain".

Response to reviewers

We thank the reviewers for their thoughtful comments. We have now revised the manuscript to systematically address their comments. A detailed response to each comment is provided below.

Reviewer #1

(Remarks to the Author): The authors provide a very detailed functional map recorded with high-density MEAs from slices of human brain organoids. They demonstrate that these slices, and thereby the organoid itself, are electrically active. Electrical activity is identified at the level of single units, of multi-unit activity and the level of local field potentials. Furthermore, the authors demonstrate that spiking activity is modulated by diazepam, and abolished by TTX, a blocker of (most) sodium channels. As the main message of the paper the authors state that “human brain organoids have self-organized neuronal assemblies of sufficient size, cellular orientation, and functional connectivity to co-activate and generate field potentials from their collective transmembrane currents that phase-lock to spiking activity.”

(1.1) Neither the technology nor the analysis tools are new. The novel aspects from a technological point are the combination of two state-of-the art electrophysiological systems with versatile computational analysis. However, many computational tools had been developed previously. Therefore, my main concern relates to the novelty of the results presented in this manuscript. To the best of my understanding, field potentials have been reported in ref.16 of this manuscript.

Response 1.1: The novel results in the paper are:

1. An analysis of ISI probability distributions suggested physiologically discrete regions of control over spiking patterns within the organoid.
2. The first comprehensive functional connectivity map within a human brain organoid. Briefly noted and further detailed in the specific responses to reviewers we define connectivity as “pairwise spike correlations” with short latencies consistent with synaptic transmission times. Importantly, connectivity does not mean a direct anatomical connection.
3. From the pairwise spike correlations we derived a graph in which edges are informative with regard to their strengths and directionality. Edge weight distributions are non-random; they are well described by a gamma distribution. We observe a relatively small set of strong pairwise spike correlations present in a large “sea” of weak correlations. This organization provides a useful way to further explore plasticity by analyzing the dynamics of edge strengths beyond the graph stability we observed within time scales of a few minutes. We further note that the majority of nodes have both incoming and outgoing spike correlations with other nodes and we refer to these nodes as “brokers.” This organization also provides a potentially dynamic parameter for the analysis of plasticity.
4. By capturing both spikes and LFPs in the same recording we demonstrated that spikes phase-lock to theta oscillations. We also provide a stronger basis for the presence of LFPs in brain organoids than is currently in the literature by demonstrating their coherence over the spatial domain of the organoid.

5. This paper is the first to demonstrate the utility of brain organoids for the analysis of CNS acting drugs in clinical use. Signature changes were observed in neuronal circuitry and network organization following treatment with diazepam that cannot be detected in any other manner.

6. While all brain organoids have issues concerning reproducibility with regard to anatomical organization and cellular composition, we show that the physiological parameters extracted in our paper are highly reproducible and constitute excellent metrics for analyzing interventions including drug treatments, mutations, and genetic engineering.

(1.2) The authors state in the second sentence of the discussion: “Over the entire area of the organoid assessed by the MEA this activity has the form of a network in which single-units are considered nodes connected by weighted directionally defined edges”. They evaluate “coupling between single-units” from spike train latency distributions (see i.e. Fig. 5b). Using this method they generate a “connectivity map”, presented i.e. in Fig.5d, 5f and suppl. Figures. The term connectivity could be misleading here. First of all, it seems as though several cell pairs are connected over an extended distance of up to 4mm, and a spike latency of about 4 ms. This very short time would be necessary to conduct action potentials along the axons. Please comment how “connected” pairs can exist over distances of several millimetres.

Response (1.2): The reviewer raises an interesting point which we need to clarify. As pointed out by this reviewer, some short latency pairwise spike correlations could result from axonal action potential propagation over extended distances. However, our analytical approach (see Methods) outlines the pipeline for determining statistically significant short-latency interactions between neurons. The high-degree of channel redundancy per neuron of our recording system facilitates the tracking and removal of the spatiotemporal footprint of single neuron spikes, including action potential propagation. We used a spike sorting method optimized for high-density field recordings (kilosort2) that removes redundant, high fidelity signals. In the example cited by the reviewer, the distances over which axons extend by immunohistochemistry are different from the distances we presented in the connectivity map. We do not observe axons tracking along the surface of our arrays at millimeter length scales (see Fig 1b,c); most likely axons extend away from electrodes into the z-plane of the organoid. Most of the distances in the connectivity map are much shorter than 4 mm and therefore latencies of ~4 ms are more feasible for processes like synaptic integration among small groups of neurons.

We have clarified our use of the term connectivity by defining its usage as “pairwise spike correlations” through which we derived an activity map or a functional connectivity map.

(1.3) An alternative explanation for the observed pattern may be one (or more) so-called central pattern generators. Pattern generator cells may be connected to many cells but not recorded here. Could the authors exclude such scenario?

Response (1.3): We cannot exclude the possibility that ‘pattern generator’ cells, or intermediate cells, possibly located in the z plane, represent an undetected shared source for the signals from which we have significant spike pairs.

(1.4) The authors evaluate theta rhythms and their phase locking to single-unit spiking. The authors need to flesh out this interesting result. Are the theta rhythms stationary or do they

move / propagate (see i.e. the study by Ferrea, Maccione et al. 2012 or by Menzler&Zeck, 2011).

Response (1.4): We observed non-stationary theta rhythms that propagate. The propagating signals are denoted by the relative phase delays as shown in Fig 7e,f which highlight a transient propagation of theta waves moving across the outer peripheral ring of the organoid slice (determined by signal averaging with respect to the reference signal denoted by the reference site 1 in Fig 7d). Panel e shows signals are correlated over a window of ~500 ms (~ 2 Theta cycles). The relative phase shifts of this signal propagation are shown in panel f denoted at two time points (t_0 and $t_0 + 60$ ms) relative to the peak signal at reference site 1. The propagating signals are also evident in Supplementary Fig 18.

To further validate the non-stationarity of the theta oscillation we computed the imaginary coherence, a technique used in EEG and MEG analysis to extract phase-shifts in LFP data (Nolte et al. *Clin Neurophysiol* 2004). This analytic approach (Suppl Fig 19) also identified the same spatial regions identified by cross correlation analysis exhibiting spatiotemporal phase shifts in theta across the organoid.

(1.5) Theta rhythms are compared to MUA but not to single-unit spiking. Any comments, why? Appropriate blocking experiments (i.e. with low dose of TTX) may inhibit spiking but not LFPs. Could the authors draw a conclusion, if the LFPs persist without (most of the) spiking?

Response (1.5): We focused on MUA because compared to single-unit activity, MUA provided a more stable temporal reference point for signal averaging theta rhythms. This may bias the MUA population rate towards neurons with a larger spatial footprint and hence channel redundancy; however, this approach has been effectively utilized for comparing population level spiking to LFP data (Whittingstal *et al.*, *Neuron* 2009). We reported single-unit spiking in relation to theta rhythms and found a subset of neurons were phase-locked to the oscillations (see Fig 8 and section titled 'Spike phase-locking to theta').

To address the point of the relationship between spiking frequency and LFPs, we demonstrated that blocking TTX-sensitive sodium channels revealed that the LFP is not distinguishable above background electronic noise (Suppl. Fig 16) consistent with the idea that the dominant contribution to LFPs are the ensemble of transmembrane currents initiated by concerted spiking (Buzsaki *et al.* *Nat Rev Neurosci* 2012). More directly to the reviewer's point, outside of the

population bursts, LFP amplitudes were dramatically attenuated during epochs of decreased spiking (suppl Fig. 17 shown to the right) pointing to a relation between the spiking and the LFP. Furthermore, reducing the number of spikes per burst as well as the burst duration with diazepam reduced spatial coherence of theta oscillations (Suppl Fig 19,20).

(1.6) The presented spike maps are presented for slices of a human brain organoid not the organoid itself. The difference should be made clear already in the header of the subsection.

Response (1.6): We thank the referee for pointing this out and have adjusted the header of the subsection.

(1.7) The authors present data from 6 organoid slices. It would be valuable to mention, how many slices could be recorded per organoid.

Response (1.7): We have added these data.

(1.8) Spike phase-locking to theta – The spikes from several electrodes seem to be locked to theta but with various time lags ? What are the implications ?

Response (1.8): Indeed, we can only speculate on the implications. Various time lags have been implicated in timing cell-specific neurons to the LFP as reported *in vivo* (Buzsaki Curr Opin Neurobiol 1995, Klausberger et al Nature 2003, Klausberger J Neurosci 2005). The detection of *in vivo* structural properties in an organoid can only be attributed to an intrinsic organization onto which *in vivo* phenomena might be mapped. We now mention these implications.

Reviewer #2

(Remarks to the Author):

The manuscript entitled "Human brain organoid networks" by Sharf and coworkers deals with the recordings of $n = 6$ brain organoids by means of high-density CMOS-based MEAs. The authors exploited the potentiality of the recording systems to evaluate both spiking and LFP activity to extract both dynamical and functional properties of the network. In addition, in $n = 4$ organoids, the authors applied a pharmacological protocol by delivering diazepam to evaluate variations in the recorded patterns of electrophysiological activity. Finally, $n = 3$ brain organoids have been used for acute recordings by combining the MEA with the Neuropixel shank in order to evaluate electrophysiological activity also in the third dimension.

Indeed, the topic of the manuscript is of great interest for the community of (computational) neuroscientists as well as for the small community of MEA users since it shows the possibility to use such devices to characterize an experimental model that is (without any doubts) more realistic than slices or cell cultures (also in 3D configuration).

However, I found the manuscript not well organized and difficult to follow, starting from the title.

We have substituted a more descriptive title using appropriate key words: "Functional Neuronal Circuitry and Oscillatory Dynamics in Human Brain Organoids"

Such manuscript seems a collection of nice and interesting results but without an organized pathway.

We have now extensively edited the introduction and structured the paper so that the logical flow and organization have been made clearer.

Here below, I will list some of the main concerns I found after a careful reading.

(2.1) The title is too vague and does not transmit any useful message. It completely loses relevant keywords like "connectivity", "spiking", "LFP", "oscillations", "propagation", "MEA". In addition, the title of the Supplementary Information file is different: "Intrinsic network activity in human brain organoids". Why?

Response (2.2): We have changed the title to: "Functional Neuronal Circuitry and Oscillatory Dynamics in Human Brain Organoids" The title error in the Supplemental materials has been corrected.

(2.2) Pg. 5. Line 94. "'Slicing of organoids....". If I understood well, brain organoids have been sliced and then coupled to MEA in order to enhance the possibility to survive. But, in terms of dynamics, how does change the activity of "an entire organoid"? did the authors perform any attempts? Consequently, which are the differences with respect to an organotypic slice? The authors should provide evidences that slices, sliced-organoid, and entire organoid display similar/dissimilar patterns of activity. This methodological approach should be also discussed in the Discussion section.

Response (2.2): We have changed the wording in our manuscript to make clear distinctions regarding when we record from organoid slices or whole organoids. We further have clarified that organoid slices are subsequently coupled to electrode arrays for extended periods. We avoid the use of the term 'organotypic slices' as such slices typically result from non-human material. Our measurements on whole organoids were presented using Neuropixels which similarly revealed neuronal spiking activity, synchronized population bursts and LFPs as measured in the slices. Our updated Results and Discussion section reflect these changes.

(2.3) Figure 1A shows an example of spiking activity map coming from a representative organoid. How do the authors explain that the activity is recorded only at the periphery of the organoid? Did they investigate the coupling in the middle of the slice? Is it possible (and which solutions) to increase the sealing (e.g., by changing the adhesion factors). In my opinion this is a strong limit of the approach, especially (see Figure 5d) if one of the aim is extract the functional connectivity map of (a layer) of the organoid. How many short connections are lost (I am referring to the example of Figure 5d).

Response (2.3): The center region of many organoids are typically composed non-neuronal, non-excitable cells (see Fig 2a,b) including cycling progenitor cells. Additionally, because our organoids are not vascularized some of the region in the middle of the slice may have experienced some cell loss due to hypoxic conditions (Fig 2e) and to lack of nutrient perfusion.

The absence of neurons in the center demonstrated immunohistochemically indicates that very little coupling is present in the middle of the slice. Our immunohistochemical data shows that the organoid slice periphery has the highest density of neuronal cells resulting in the peripheral ring of activity observed in Fig 1a. This organization was further revealed when inserting the Neuropixels probe into whole organoids (Fig. 9b), which also showed the absence of activity in the central region of the organoid. Some organoid slices do have spiking in the center and periphery (Suppl Fig 4), and these organoids generate similar data among all the parameters measured here in terms of LFPs, ISI distributions and network formation. It might also be mentioned that cavitation or encephalomalacia occurs in living human brains, often after injury or stroke, and the brain continues to function by re-wiring around the injury.

(2.4) I found a little bit far from the topic o the work, the rigorous characterization of the neuronal composition o the organoid. This is not a novel result. The authors could cite many o the works of the literature that have already detected the different neuronal types. Although nice, is Figure 2 fundamental?

Response (2.4): Some methods of organoid preparation fail to produce significant numbers of inhibitory neurons, particularly parvalbumin-positive neurons. Figure 2 supports the fact that the organoids we used have the full complement of cell types described in other reports, including parvalbumin-positive neurons and other inhibitory neurons known to contribute to LFPs. Thus, the cell composition we demonstrated is intended to show that our system is capable of supporting the physiological observations we report.

(2.5) Since the analyses have been performed with a small sample o organoids, statistics analyses have ben often not done. And this is not a problem, and I found much more informative to display the metrics organoid by organoid. What I do not like, that I in not rigorous is the use of the confidence level instead of the p-value in some analysis. For example line 225, pg 10. I would see the p-values.

Response (2.5): We have presented p-values as suggested.

(2.6) At the beginning of the Section "Short-term interactions in human brain organoids", the authors should better explain the meaning of "short-terms interactions" (line 296), by introducing (for example) time constants that can also sustain the choice of the used correlation parameters.

Response (2.6): We have changed the terminology "short term interactions" to short latency interactions to more accurately define our observations. This term refers to robust interactions with significant, short-latency peaks (≈ 5 ms) in the cross-correlation of action potentials between neuron pairs. This original nomenclature and technique was used by Bartho and Buzsaki *J Neurophysiol* 2004 to identify connections between neurons *in vivo* using high-density extracellular recordings. However, we agree with the reviewer that our revised term is preferred.

(2.7) Regarding the correlation analysis, how long are the used recordings? are the achieve results stable? i the authors try to split the recordings to verify the stability of the correlation outcomes? Please, discuss!

Response (2.7): Additional analyses were performed based on the reviewer's suggestions. Organoid recordings (three minute durations) were split in half and subjected to the same pairwise spike correlation analysis and shown in Suppl. Fig.13c (see figure below). The distribution of correlation strengths between units did not vary significantly (determined by a two-sample KS test; $P > 0.1$) when comparing the first 90 seconds of the recording to the second 90 seconds of the recording. These findings are also consistent with the same set of active single-units as well as pairwise spike correlations that also did not significantly vary over the course of a 4-hour time window (Suppl. Fig. 13a-b). We have now included these important points in the updated manuscript based on the reviewer's excellent suggestions.

(2.8) Figure 5c is not fundamental, and can be included as subplot.

Response (2.8): Figure 5c is now shown as a subplot

(2.9) The last statement at pg 14 is weak. I suggest to remove.

Response (2.9): We have removed the last statement as suggested.

(2.10) I found very interesting the analysis on the kind of nodes (sender, receiver, broker), but I think that it should be better discussed. Do the authors have an idea of the achieved percentage? How did they choose the threshold 0.8? How robust is such a value?

Response (2.10): We thank the reviewer noting these observations. We presented the percentage of sender, receiver and brokers on p.16 line 348 as well as Supplemental Fig 12a (please see below).

Nodes with a high fraction of incoming edges $(D_{in} - D_{out}) / (D_{out} + D_{in}) > 0.8$ were labelled 'receiver' nodes (Fig. 5e, middle). Differences in the directionality vector less than 0.8 were labelled 'broker' nodes (Fig. 5d, bottom), which represented the majority of the nodes (senders: $15.4\% \pm 2.6\%$, brokers: $62.7\% \pm 6.0\%$, receivers: $21.9\% \pm 6.5\%$ (mean \pm STD)). These data were computed from four independent organoids (Supplementary Fig. 12b).

The updated Supplementary Fig 12 addresses the issue of the 0.8 threshold. This threshold, for which at least 90% of the edges are outgoing or incoming (which results in the 0.8 cutoff value), are chosen as senders/receivers. As shown in panel a (reproduced below) the relative fraction of nodes with primarily incoming or outgoing edges shows a sharp increase for $|(D_{in} - D_{out}) / (D_{out} + D_{in})| > 0.9$. Randomizing the pairwise correlation matrix (performing degree preserving double edge swaps) yielded no sender or receiver nodes (panel b). The manuscript now includes these additional updates.

(2.11) The sentence "This result was consistent..." (line 360-362) is weak. I would remove.

Response (2.10): The sentence has been removed.

(2.12) Regarding the "Acute measurements in whole organoids" section, what I think it could be disruptive is the simultaneous recording of the brain slice from MEA (2D) and by the Neuropixel (3D) and then try to extract 3D features.

In 2021, Shin and coworkers published a great work on Nat. Comm where they recorded the activity by means of 3D high-density MEAs and were able to reconstruct the 3D functional connections of the network.

Instead, in this manuscript, the authors only exploited the presence of the shank to evaluate whether there are similar electrophysiological features of the 2D layer. I know that new experiments are time-consuming, but why the authors do not organize an experimental

campaign where CMOS-MEA and Neuropixel are simultaneously used?

Response (2.12): We thank the reviewer for pointing out Shin et al., 2021 where the authors used 3D HD-MEAs to reconstruct 3D functional connections. We now cite this work in our paper, the technological innovations pioneered in their paper are extremely relevant to our organoid work. Developing a system to simultaneously record from the CMOS-MEA and Neuropixels shanks would require substantial engineering and design to integrate the two systems together in an environment amenable for prolonged slice viability and is thus beyond the scope of our initial manuscript. We are however inspired by the work of Shin and coworkers and are eager to engineer such a system for future organoid studies.

(2.13) The discussion section should be completely revised. Not all the figures are critically discussed as well as the results are poorly discussed with respect to the state of the art. Which is the most novel thing that should emerge from this manuscript? In the current version of the manuscript, I have difficulties to find it.

Response (2.13): The discussion has been completely rewritten to address the figures critically, to provide some further interpretation of the data and to emphasize more clearly the novel aspects of our work as bulleted in response to reviewer #1.

(2.14) Data and materials availability do not satisfy the current standard. In 2021, it is unacceptable to read "Additional data and code related to this paper are available from the corresponding author upon reasonable request". Please, share data and code with a doi in public repository.

Response (2.14): We have updated the Data and materials availability (reproduced below).

Data Availability: The data support the findings of this study are available within the article and its supplementary information files and for the very large raw data sets, from the corresponding author upon request. Source data are provided with this paper.

Code Availability: Spike sorting was performed in Python 3.6 using SpikeInterface 0.13.0 and previously published (Buccino *et al.* *eLife* 2020), which can be found here <https://github.com/SpikeInterface/spikeinterface>. Pairwise spike correlation analysis was performed in MATLAB (version 2018b) utilizing previously published code in C (Cutts *J Neurosci* 2014) that has been adapted to MATLAB (Giandomenico *et al.* *Nat Neurosci* 2019) and can be found here <https://github.com/Timothysit/organoids>. Custom code for the visualization of organoid network activity is available at <https://github.com/KosikOrganoid/Intrinsic-activity-code>.

Reviewer #3

(Remarks to the Author):

Sharf et al. have produced a remarkable characterization of neuronal activity in human brain organoids using CMOS-based microelectrode array (MEA) comprised of 26,400 electrodes in the x-y plane and a 960 electrode Neuropixels CMOS shank probe in the z-direction. The conclusions are comparable to those put forth in a recently published article (PMID 34426698),

i.e., human brain organoids with excitatory and inhibitory neurons are capable of generating complex oscillatory activities not unlike in the intact brain.

We thank the reviewer for comparing our work on organoid signaling to the recently published findings of Samarasinghe *et al.* 2021 (ref.10). However, we would like to highlight a few key differences between our work and theirs. (1) Samarasinghe *et al.* inserted a patch pipette into their organoids and recorded field potentials from highly localized regions in the tissue. Our work reveals that spatiotemporal components of LFP oscillations are not homogeneously distributed across the organoid, an organoid feature that cannot be obtained using a single patch pipette. (2) Our work simultaneously captures LFPs and spiking activity whereas Samarasinghe *et al.* relied on calcium signaling which cannot resolve action potentials to access network activity in their organoids. Our conclusions regarding spiking and LFPs were readily detected with the high density electrode platform.

Instead of using a disease model, the authors of the present study carried out sophisticated analyses for determining connectivity within the organoid and establishing phase-locked firing of the neurons with theta frequency oscillations. I have a few comments that should be considered for a better documentation of the neuronal activity that takes place in the organoids.

(3.1) The authors describe the firing of the neurons in the organoids as “bursts”. However, looking at the samples of the firing patterns it is evident that the firing is much more complicated than just simple bursts. There seem to be bursts of bursts (clusters) and perhaps even bursts of clusters (superclusters). This nomenclature comes from the single-channel literature, where the patterns of channel openings and closures determine the kinetic schemes assigned to channel activity. I recommend using the log-binned plots of the inter-spike intervals (ISI), as the closed times are plotted for single channels, to determine the nested exponential distributions of the ISIs (e.g., see PMID 6131450). This would be a better graphical representation than the cumulative probability distributions plotted on a linear ordinate (e.g., Fig.3 a & b), where it appears that about 20% of the spikes have 0 Hz firing rate, which is impossible. If the exponential distributions of the ISI hold, then there will be no need for artificially fitting the cumulative Weibull distributions, as these distributions do not seem to fit too well, as illustrated on Fig 3a both for the control and diazepam conditions.

Response (3.1): These points have been exceptionally helpful. We have noted that what are called bursts are indeed considerably more complicated when the detailed firing patterns within bursts are analyzed. The suggested analysis of the ISIs led us toward an additional novel insight, specifically the locality of ISI probability distributions (Fig 3) consistent with primate recordings (Barbieri, R., Quirk, M. C., Frank, L. M., Wilson, M. A. & Brown, E. N. Construction and analysis of non-Poisson stimulus-response models of neural spiking activity. *J. Neurosci. Methods* **105**, 25–37 (2001); Maimon, G. & Assad, J. A. Beyond Poisson: Increased Spike-Time Regularity across Primate Parietal Cortex. *Neuron* **62**, 426–440 (2009); Mochizuki, Y. *et al.* Similarity in neuronal firing regimes across mammalian species. *J. Neurosci.* **36**, 5736–5747 (2016)). Comparing our single-unit ISI histograms to an exponentially distributed ISI probability density (characteristic of an ideal Poisson spike train) we can quantify the degree of

randomness of single-unit firing patterns. Further, we have simplified the analysis, utilize a simple coefficient of variation (ratio of standard deviation to mean) to convey the extent of ISI regularity and irregularity by deviation from unity (expected with a Poisson). We have discarded the Weibull distributions.

(3.2) The authors make a good point about the theta frequency oscillations being phase locked to the firing of action potentials. However, on the raw LFP traces there appear to be higher frequency oscillation occurring together with the theta rhythms (e.g., Fig. 7a). Was there any phase-amplitude or frequency-amplitude coupling between low and high frequency oscillations? This would be a nice further demonstration of highly organized network activity being present in the organoids.

Response (3.2): We did observe phase-amplitude coupling between low- and high-frequency oscillations, for example, high-gamma frequency (100-400 Hz) amplitude coupling to a range of delta frequency (0.5-4 Hz) phases across organoids. Additionally, we have LFP oscillation amplitudes over a range of different sub-bands (see spatial maps in Suppl. Fig. 15). We consider this topic quite extensive and therefore should be the topic of a separate manuscript in order not to distract from the main narrative here.

(3.3) It is regrettable that the histology and the recordings are shown on different organoid cross-sections. It would be extremely informative to present the histology even with just a pan-neuronal marker such as NeuN for the electrophysiological map shown in Fig.1a.

Response (3.3): We agree that a histology map of a pan neuronal marker for the electrophysiology map would be informative. However, when we attempted to remove organoid sections attached to CMOS arrays, the tissue would not detach without aggressive enzymatic digestion, causing significant tissue damage. The attachment of tissue to the CMOS surface results from the micron-scale variation in the CMOS recording surface designed to provide a large surface area for cell attachment (see image of array surface morphology below, courtesy of Maxwell Biosystems) and thus may intrinsically limit the ability to correlate morphology with physiology.

(3.4) This is a minor point, but it is highly unlikely that people with color vision deficiency will get much out from the color scales for the phase in several figures (e.g., Fig. 7f).

Response (3.4): A verbal description of the conclusions of the color scales has been added as a Suppl legend for Figs 7,8,9 as well as grayscale images of the figure panels.

Supplementary Legend 1 for Fig. 7f,i. Left panel, the inner contour of solid circles have phases ~120 degrees. The pocket of circles on the center right consists of phases ~220-240 degrees. The bottom left pocket consists primarily of phases ~330-360 degrees. Right panel, the phases are advanced by ~100 degrees. Grayscale images are shown below.

Supplementary Legend 2 for Fig. 8c. The cluster of circles directly above marker 1 ($\mu = 103$ degrees) have averages phases ~ 300 degrees (two circles on the bottom left of the cluster have phases ~ 60 and 190 degrees), while the circles neighboring marker 2 ($\mu = 189$ degrees) and marker 3 ($\mu = 16$ degrees) have phases ~ 200 degrees.

Supplementary Legend 3 for Fig. 9e. The circles in the upper half of the two columns have a predominate phase of ~160 degrees. The larger two dark circles in the center of the left column have phases ~190 degrees. Further down the left column, as the circles diminish in size their phase also decreases to ~160 degrees, and as the circles increase in size their phase advances to ~190 degrees. The larger circles in the center half of the right column are ~170 degrees. A grayscale image is shown below.

Additional clarifications have been added to the supplemental figure legends.

(3.5) The last sentence of the Discussion attributes too much to the activity recorded in the sensory input and motor output deprived organoids. After all, Sherrington stated that “The brain seems a thoroughfare for nerve-action passing its way to the motor animal”. Accordingly, without a motor output, the organoids should not even be denoted as “brain”.

Response (3.5): Thank-you for reminding us of Sherrington’s provocative quote. It is certainly not our intention to attribute too much significance to the organoid’s waveforms. Indeed, the word brain even when qualified as an organoid probably over-estimates these tissue cultures. But brain organoid is a welcome advance over the previous term, “minibrain.”

Reviewers' Comments:

Reviewer #1:

Remarks to the Author:

The authors replied to all comments and revised the manuscript substantially. Indeed, the discussion section has been rewritten and now constitutes one of the best sections of the entire manuscript. Some of my concerns regarding the novelty and scientific rigor, however, have not been addressed. I kindly ask the authors to clarify their results (i.e. provide quantifications).

Main criticism:

- The result section remains vague, with little quantitative information. Many sentences refer to subplots without concluding with a quantitative result. For example in line 415, "...(we) found sets of highly coherent electrodes (Supplementary Fig. 19), which corresponded to the same spatial regions identified by cross-correlation and signal averaged analysis". However, the spatial regions are difficult to identify in figure 7f and 7i and the authors did not provide any quantification of the "spatial region".
- Given that the authors emphasize the "coherence over the spatial domain of the organoid" as one of the novel aspects of their paper I am still puzzled how they infer this result.

- The authors claim to show in panel 7c a "zoomed view of highlighted black rectangle in b". The short sequence in (b) looks different, however.

- Phase-locking of spikes to LFPs. In figure 8c about 20 units are shown to phase-lock to the LFP. However, at least the same number (shown in grey) don't phase-lock. I could not find any quantitative estimation of the degree of phase-locking.
- Furthermore, the authors claim (line 475): "Previous work in vivo and ex vivo has relied on manually positioned, low-density recording electrodes, to identify a handful of units exhibiting preferential spike phase-locking to theta frequencies at a given moment in time [52,53,55,56]". This statement is wrong and I provided the authors at least two references in my first review, where high-density electrode arrays have been used to demonstrate spike phase locking to LFPs.
- I was puzzled to find only in the discussion but not in the result section the number of identified cells (line 564: "a set of 224 neurons analyzed from four different organoids"). I encourage the authors to provide this quantity earlier in the manuscript.

Reviewer #2:

Remarks to the Author:

The authors made a good job to improve the quality of their manuscript. However I continue to have strong concerns about (especially) the relevance of this work. Here below (by referring to the authors' answers), I listed my observations.

1. Response 2.2  I appreciated the more clarity to explain that sliced-organoid were coupled to MEAs. However, I do not find this approach elegant and able to furnish a valid experimental model. In addition, the authors did not provide any comparison of recordings between sliced organoids and entire organoids (coupled to a MEA). Such an experimental approach is (in my opinion) a strong limitation. Also because the time the experimenter needs to perform a recording is very long. Why don't using neural spheroids? See Pasca's lab for example.

2. Response 2.12 - The interfacing of CMOS-MEA and Neuropixel would be the very novelty of this work, with a strong engineering component together with the computational one to analyze data and map simultaneously the electrophysiological activity of a complex neuronal structure. Honestly, the solution provided by the authors is an oversimplification.

3. Response (2.14) - In 2022, this kind of response would deserve immediate rejection. All the

data should be shared! The entire dataset! The authors should think about the many repositories available. For the best of my knowledge, EU projects (and in the Ack section I read some of these) require dataset with their own doi.

Reviewer #3:

Remarks to the Author:

I am satisfied with the authors having addressed my comments and suggestions.

Response to reviewers

We thank the reviewers for their patience and further suggestions to improve the quantitative data presentation and clarify the novelty of this research. The manuscript has been revised to address all of their comments in detail.

Reviewer #1

(Remarks to the Author): The authors replied to all comments and revised the manuscript substantially. Indeed, the discussion section has been rewritten and now constitutes one of the best sections of the entire manuscript.

Thank-you.

Some of my concerns regarding the novelty and scientific rigor, however, have not been addressed. I kindly ask the authors to clarify their results (i.e. provide quantifications).

We have provided the quantifications as suggested by the reviewer. We have also clarified novel aspects of our findings in the context of the organoid field as well as the broader implications of our work in relation to observations found in *in vivo* brain networks.

(1.1) The result section remains vague, with little quantitative information. Many sentences refer to subplots without concluding with a quantitative result. For example in line 415, "... (we) found sets of highly coherent electrodes (Supplementary Fig. 19), which corresponded to the same spatial regions identified by cross-correlation and signal averaged analysis". However, the spatial regions are difficult to identify in figure 7f and 7i and the authors did not provide any quantification of the "spatial region".

Response 1.1: We thank the reviewer for directing us toward improved identification of the spatial regions mentioned in Figure 7 identified by cross-correlation (Fig 7f,i) and the imaginary coherence analysis (Suppl. Fig. 19). We have now provided additional quantifications to make this overlap more apparent (Suppl. Fig. 19g,h). The overlap between the most highly correlated imaginary coherence cluster (top 67%, red circles) and the sites showing the highest cross correlation (top 18% based on correlation threshold, black open circles) are shown in panel g below, yielding a 44% overlap. Furthermore, we now calculate the overlap fraction as a function of correlation threshold for both the cross-correlation and the imaginary coherence estimates. The thresholds used in panel g are indicated by the arrow in panel h. We replicated the same quantitative analysis in another organoid and observed similar overlap at the same threshold values (51% percent overlap)

(1.2) Given that the authors emphasize the “coherence over the spatial domain of the organoid” as one of the novel aspects of their paper I am still puzzled how they infer this result.

Response (1.2): Thank-you for requesting this important clarification. We now address critical aspects of stationary vs. propagating aspects of LFPs (previously mentioned by the reviewer) that have been elegantly mapped using HD-CMOS arrays by Ferrea, Maccione et al. 2012 (hippocampal-cortical slices) and by Menzler and Zeck 2011 (retinal preparations). We now cite these publications. We have performed additional analyses (now included additional supplemental figures and shown below) that demonstrate non-stationary features of the LFP in our organoids to further complement our spatial correlation analyses. Our discussion has now been updated to include the implications of non-stationary components of the LFP in organoids and how that relates to propagating patterns that may reflect anatomical organization *in vivo*.

(1.3) The authors claim to show in panel 7c a “zoomed view of highlighted black rectangle in b”. The short sequence in (b) looks different, however.

Response (1.3): We have provided the diagram below with arrows to illustrate that the rectangle in 7b is the same as the zoomed in view in panel 7c. However, the gray rectangle in panel 7b was manually drawn and should not have extended all the way to the right end of the axis in panel 7b. We have corrected this to make interpreting the figure easier in the revised manuscript.

(1.4) Phase-locking of spikes to LFPs. In figure 8c about 20 units are shown to phase-lock to the LFP. However, at least the same number (shown in grey) don't phase-lock. I could not find any quantitative estimation of the degree of phase-locking.

Response (1.4): The manuscript has been updated to include these details.

(1.5) Furthermore, the authors claim (line 475): "Previous work in vivo and ex vivo has relied on manually positioned, low-density recording electrodes, to identify a handful of units exhibiting preferential spike phase-locking to theta frequencies at a given moment in time [52,53,55,56]". This statement is wrong and I provided the authors at least two references in my first review, where high-density electrode arrays have been used to demonstrate spike phase locking to LFPs.

Response (1.5): We thank the reviewer for reminding us of these key papers performed using HD-CMOS arrays by Ferrea, Maccione et al. 2012 (hippocampal-cortical slices) and by Menzler and Zeck 2011 (retinal preparations). Indeed, the authors demonstrate that spikes are strongly correlated during epochs of increased LFP amplitude. In those examples, tissue was grown in an animal from well-defined anatomical structures. We now discuss the implications of phase-locking that arise in a self-organized system derived from human iPSCs.

(1.6) I was puzzled to find only in the discussion but not in the result section the number of identified cells (line 564: "a set of 224 neurons analyzed from four different organoids"). I encourage the authors to provide this quantity earlier in the manuscript.

Response (1.6): We agree, these details belong in the results section and have put them there.

Reviewer #2

(Remarks to the Author):

The authors made a good job to improve the quality of their manuscript. However I continue to have strong concerns about (especially) the relevance of this work. Here below (by referring to the authors' answers), I listed my observations.

We thank the reviewer for these comments and suggestions to improve the quality of our work.

(2.1) Response 2.2  I appreciated the more clarity to explain that sliced-organoid were coupled to MEAs. However, I do not find this approach elegant and able to furnish a valid experimental model. In addition, the authors did not provide any comparison of recordings between sliced organoids and entire organoids (coupled to a MEA). Such an experimental approach is (in my opinion) a strong limitation. Also because the time the experimenter needs to perform a recording is very long. Why don't using neural spheroids? See Pasca's lab for example.

Response (2.1): The last section of manuscript is dedicated to recordings from whole organoids. Furthermore, we demonstrate that several key features are preserved in whole organoid recordings. (1) Neuronal population bursts occur as observed in our slice preparations over similar timescales; (2) LFP oscillations occur in the theta frequency range; (3) Neuronal population bursts are also synchronized with the LFP; (4) single neuron firing patterns demonstrate phase-locking to theta oscillations.

The discussion section has been updated to highlight advantages and limitations of organoid slice models as a platform to more faithfully recapitulate features of the brain cytoarchitecture found *in vivo*. Finally, we now further discuss the tradeoffs between measuring activity from slices compared to intact signals obtained in animal models.

We are well aware of the enormous contributions from the Pasca lab; however, there is no *a priori* reason to prefer the Pasca methods in favor of the studies we presented—both methods have utility and we provide extensive documentation for the cellular composition and tissue organization of the organoids we used. Finally, both organoid model systems follow a similar developmental trajectory, requiring many months to mature and generate a diverse range of neuronal cell types.

(2.2) Response 2.12 - The interfacing of CMOS-MEA and Neuropixel would be the very novelty of this work, with a strong engineering component together with the computational one to analyze data and map simultaneously the electrophysiological activity of a complex neuronal structure. Honestly, the solution provided by the authors is an oversimplification.

Response (2.2): We believe the novel aspects of our work are not the engineering aspects, but rather, using state-of-the-art MEAs to reveal details of neuronal network activity that emerge in a brain organoid.

Combining two separate commercial probe technologies in the same organoid would be a major investment of time and resources and beyond the scope of our initial study. As mentioned by the reviewer, the work of Shin *et al.* Nat Comm 2021 was focused entirely on developing a 3D ephys measurement apparatus. We agree that engineering a system for a simultaneous 3D readout that is properly designed for long term culture viability, as needed for these measurements, would be a major undertaking and would constitute a publication in its rite.

(2.3) Response (2.14) - In 2022, this kind of response would deserve immediate rejection. All the data should be shared! The entire dataset! The authors should think about the many repositories available. For the best of my knowledge, EU projects (and in the Ack section I read some of these) require dataset with their own doi.

Response (2.3): We acknowledge this oversight regarding a public repository capable of supporting our large data sets. We have uploaded the data sets to the repository DRYAD and have updated our Data Availability section with a link to the doi.

Reviewer #3

(Remarks to the Author):

I am satisfied with the authors having addressed my comments and suggestions.

Reviewers' Comments:

Reviewer #1:

Remarks to the Author:

As I reviewed the manuscript previously two times I briefly mention here, that the authors now addressed all my comments in an appropriate way. I recommend the manuscript for publication. In my opinion it certainly represents a step forward understanding the intricate network of human brain organoids.

Response to reviewers

We thank the reviewers for their patience and further suggestions to improve the quantitative data presentation.

Reviewer #1

Reviewer #1 (Remarks to the Author):

As I reviewed the manuscript previously two times I briefly mention here, that the authors now addressed all my comments in an appropriate way. I recommend the manuscript for publication. In my opinion it certainly represents a step forward understanding the intricate network of human brain organoids.

Thank-you.